# The lineage stability and suppressive program of regulatory T cells require protein O-GlcNAcylation

Bing Liu [1], Oscar C. Salgado[2], Sangya Singh[3], Keli L. Hippen[3], Jason C. Maynard[4], Alma L. Burlingame[4], Lauren E. Ball[5], Bruce R. Blazar[3], Michael A. Farrar [2,6], Kristin A. Hogquist[2,6] & Hai-Bin Ruan [1,2]

Regulatory T (Treg) cells control self-tolerance, inflammatory responses and tissue homeostasis. In mature Treg cells, continued expression of FOXP3 maintains lineage identity, while T cell receptor (TCR) signaling and interleukin-2 (IL-2)/STAT5 activation support the suppressive effector function of Treg cells, but how these regulators synergize to control Treg cell homeostasis and function remains unclear. Here we show that TCR-activated posttranslational modification by O-linked N-Acetylglucosamine (O-GlcNAc) stabilizes FOXP3 and activates STAT5, thus integrating these critical signaling pathways. O-GlcNAc-deficient Treg cells develop normally but display modestly reduced FOXP3 expression, strongly impaired lineage stability and effector function, and ultimately fatal autoimmunity in mice. Moreover, deficiency in protein O-GlcNAcylation attenuates IL-2/STAT5 signaling, while overexpression of a constitutively active form of STAT5 partially ameliorates Treg cell dysfunction and systemic inflammation in O-GlcNAc deficient mice. Collectively, our data demonstrate that protein O-GlcNAcylation is essential for lineage stability and effector function in Treg cells.

[1] Department of Integrative Biology and Physiology, University of Minnesota, Minneapolis, MN 55455, USA. [2] Center for Immunology, University of Minnesota, Minneapolis, MN 55455, USA. [3] Division of Blood and Marrow Transplantation, Cancer Center and the Department of Pediatrics, University of Minnesota, Minneapolis, Minnesota 55455, USA. [4] Department of Pharmaceutical Chemistry, University of California, San Francisco, San Francisco, CA 94158, USA. [5] Department of Cell and Molecular Pharmacology and Experimental Therapeutics, Medical University of South Carolina, Charleston, SC 29425, USA. [6] Department of Laboratory Medicine and Pathology, University of Minnesota, Minneapolis, MN 55455, USA. These authors contributed equally: Bing Liu, Oscar C. Salgado. Correspondence and requests for materials should be addressed to K.A.H. (email: hogqu001@umn.edu) or to H.-B.R. (email: hruan@umn.edu)

Regulatory T (Treg) cells are distinct T lymphocytes that control immunological self-tolerance and homeostasis[1,2]. The lineage-defining transcription factor Forkhead box P3 (FOXP3), together with other transcription regulators, induces Treg cell development in the thymus. T-cell receptor (TCR)-derived and interleukin-2 receptor (IL-2R)-derived instructive signals act in two steps to induce the *Foxp3* gene expression in developing Treg cells[3–5]. Deleting or mutating the *Foxp3* gene leads to the scurfy phenotype characterized by multi-organ inflammation in mice[6–8]. In mature Treg cells, continued expression of FOXP3 maintains their lineage identity;[9,10] however, a small but significant population of Treg cells may lose FOXP3 expression and acquire effector T-cell activities in normal and particularly inflammatory settings[11–13]. Nevertheless, molecular mechanisms controlling FOXP3 protein stability under homeostatic and pathologic conditions are not well understood.

Effector Treg (eTreg) cells are the most biologically potent population of Treg cells[14,15]. Recent studies have demonstrated that pathways that regulate Treg cell development are also required for the formation and function of eTreg cells. Continuous TCR signaling maintains the transcriptional program and suppressive function of eTreg cells, without affecting *Foxp3* gene expression[16,17]. IL-2R and downstream STAT5 signaling are also indispensable for eTreg cell differentiation and function by controlling a distinct set of genes that are separable from those regulated by TCR signaling[18]. It is still unclear how Treg cells integrate these pathways to maintain the suppressive program.

Post-translational modification networks exist in Treg cells to rapidly integrate signals from diverse environmental stimuli to modulate Treg cell function accordingly. In this regard, the FOXP3 protein has been intensively investigated. FOXP3 can be regulated by phosphorylation, acetylation, and ubiquitination in response to environmental changes to modulate its protein stability and DNA-binding ability[19]. In recent years, a novel modification was discovered: O-linked N-Acetylglucosamine (O-GlcNAc) modifies intracellular proteins at serine and threonine residues[20]. O-GlcNAcylation is radically different from other types of glycosylation, and, analogous to phosphorylation, plays a central role in signaling pathways relevant to chronic human diseases including cardiovascular disease, diabetes, neurodegeneration, and cancer[21,22]. The enzymes O-GlcNAc transferase (OGT) and O-GlcNAcase (OGA) mediate the addition and removal of O-GlcNAc, respectively. We and others have demonstrated that O-GlcNAc signaling acts as a hormone and nutrient sensor to control many biological processes such as gene transcription, protein stability, and cell signaling[23–26].

Earlier studies have shown that T cells express and upregulate O-GlcNAcylation upon immune activation[27]. T cell-specific ablation of OGT resulted in an increase of apoptotic T cells[28], and blocked T cell progenitor renewal, malignant transformation and peripheral T cell clonal expansion[29]. These data demonstrate that protein O-GlcNAcylation links TCR signaling to T cell differentiation and function; however, the role of O-GlcNAcylation in Treg cells has not been studied.

Here, we demonstrate that protein O-GlcNAcylation is abundant, and is functionally important in Treg cells by modifying FOXP3 and STAT5. Selective ablation of OGT in Treg cells leads to an aggressive autoimmune syndrome in mice as a result of Treg lineage instability and eTreg cell deficiency. On the other hand, pharmacological elevation of protein O-GlcNAcylation enhances the suppressive activity of human Treg cells, which will provide insights to help us better manipulate these cells in patients to treat diseases such as autoimmune disorders, transplant rejection and cancer.

## Results

**FOXP3 is modified and stabilized by O-GlcNAcylation.** TCR-activated protein O-GlcNAcylation is critical for T-cell development and function[29]. We found that, similar to CD4[+]CD25[−] naïve T cells, CD4[+]CD25[+]FOXP3[+] Treg cells displayed abundant expression of OGT and global protein O-GlcNAcylation (Fig. 1a, b), implying a potential role of O-GlcNAcylation in Treg cells. Consistent with findings in T cells, TCR activation further promoted protein O-GlcNAcylation in Treg cells ex vivo (Fig. 1c). We also stimulated naïve T cells with TGFβ to generate induced Treg (iTreg) cells in vitro. Compared with cells only treated with anti-CD3/CD28 beads, iTreg cell showed increased levels of the *Ogt* gene expression and global protein O-GlcNAcylation (Supplementary Fig. 1A, B). These data indicate that TCR activates protein O-GlcNAcylation in Treg cells.

The FOXP3 protein is subjected to various posttranslational modifications that are required for lineage maintenance and suppressive function[19]. Thus, we sought to test whether the FOXP3 protein itself could be modified by O-GlcNAcylation. FOXP3 O-GlcNAcylation could be detected when ectopically expressed in human embryonic kidney (HEK) 293 cells (Fig. 1d). OGT overexpression increased levels of total and O-GlcNAcylated FOXP3, while OGA decreased FOXP3 protein expression and its O-GlcNAcylation (Fig. 1d). Similarly, levels of total and O-GlcNAcylated FOXP3 were increased when OGA was inhibited by Thiamet-G (TMG); in contrast, inhibition of OGT by ST045849 decreased both total and O-GlcNAcylated FOXP3 (Fig. 1e). FOXP3 protein degradation induced by the protein synthesis inhibitor cycloheximide (CHX) could be prevented by the proteasome inhibitor MG132 (Fig. 1f), indicating that FOXP3 was degraded through a ubiquitin/proteasome-dependent pathway. OGT overexpression or OGA inhibition increased FOXP3 stability (Fig. 1f-h), while OGA overexpression or OGT inhibition destabilized FOXP3 (Fig. 1g, i). The ubiquitin ligase STUB1 and the deubiquitinase USP7 have been reported to control FOXP3 polyubiquitination and degradation[30,31]. STUB1 overexpression reduced FOXP3 levels, which could not be further decreased by OGA. When USP7 was present, OGT could not further increase FOXP3 protein levels (Supplementary Fig. 1C). These data suggest that O-GlcNAcylation may counteract ubiquitination to stabilize FOXP3 protein.

We then sought to determine whether O-GlcNAcylation controls FOXP3 protein stability in Treg cells. CD4[+]CD25[+] Treg cells from inducible OGT knockout (KO) mice (*Ubc-Cre/ERT2[+]Ogt[fl/Y]*) were isolated and expanded ex vivo in the presence of anti-CD3/CD28 antibodies and recombinant IL-2. 4-Hydroxytamoxifen (4-OHT) treatment in wildtype cells had no adverse effect on protein O-GlcNAcylation, FOXP3 expression, or Treg cell number (Supplementary Fig. 2A-C). In *Ubc-Cre/ERT2[+]Ogt[fl/Y]* cells, however, 4-OHT reduced global O-GlcNAcylation (Supplementary Fig. 2D), FOXP3 protein abundance on a per-cell basis (Fig. 2a, b), and frequencies of CD4[+]CD25[+] (Supplementary Fig. 2E) and CD4[+]FOXP3[+] Treg cells (Fig. 2c, d), demonstrating that loss of O-GlcNAcylation destabilizes FOXP3 protein in Treg cells ex vivo.

To identify sites of FOXP3 O-GlcNAcylation, Flag-tagged FOXP3 was expressed in HEK 293 cells together with OGT, immunopurified with anti-Flag beads, trypsin digested, and analyzed by liquid chromatography with tandem mass spectrometry (LC-MS/MS) using electron transfer dissociation (ETD). Multiple O-GlcNAcylation sites were identified (Supplemental Data 1). Mutating 5 of these sites, including Thr38, Ser57, Ser58, Ser270, and Ser273, to alanine (5A) on the FOXP3 protein significantly blunted its O-GlcNAcylation level (Fig. 2e), reduced its stability (Fig. 2f), and ablated its transcriptional suppression activity induced by OGT (Supplementary Fig. 2F). We then

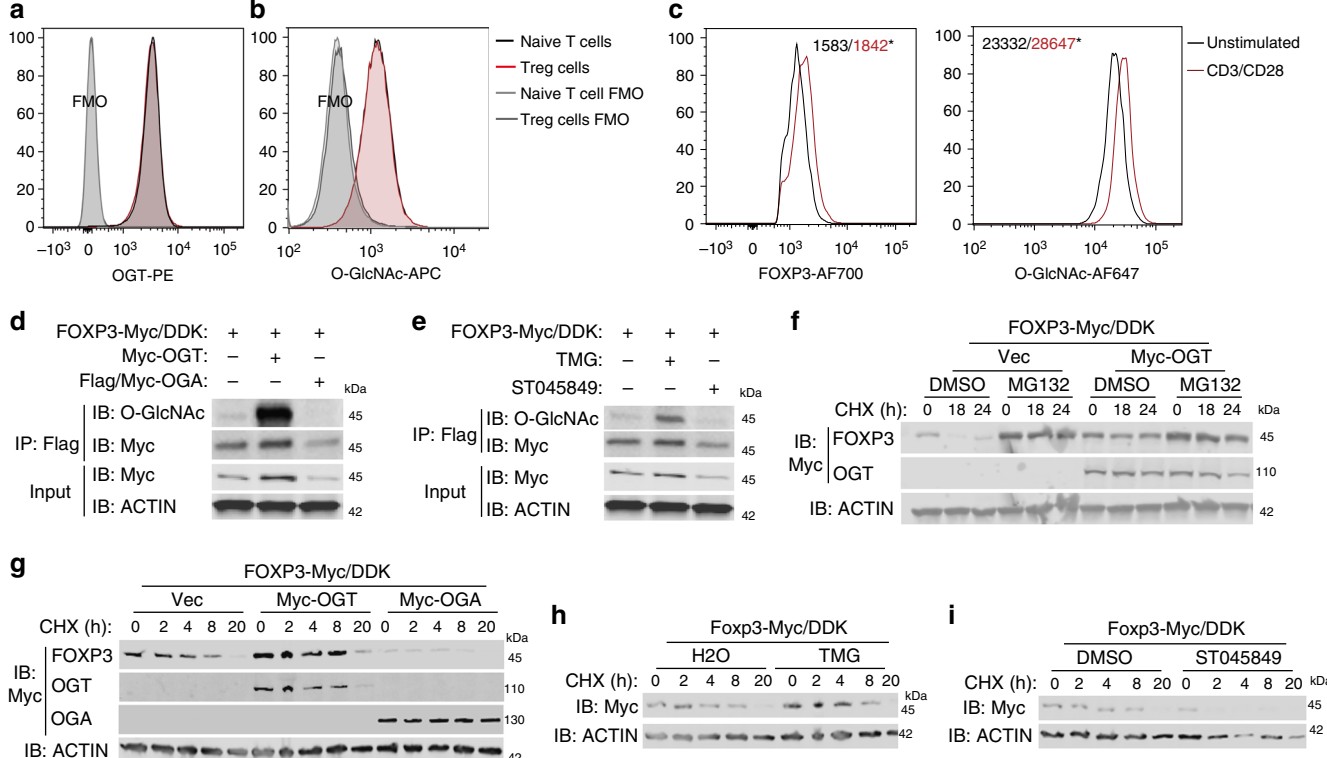

**Fig. 1** O-GlcNAc-cycling enzymes regulate FOXP3 stability in vitro. **a**, **b** Mean fluorescence intensity (MFI) of OGT (**a**) and O-GlcNAcylation (**b**) in CD4+CD25−naïve T cells, CD4+CD25+FOXP3+ Treg cells and corresponding Fluorescence Minus One (FMO) negative controls. **c** Treg cells isolated from wildtype mice were stimulated with or without anti-CD3/CD28 beads for 24 h ex vivo (n = 3). MFI of FOXP3 and O-GlcNAcylation was analyzed in CD4+CD25+ FOXP3+ Treg cells. **d**, **e** HEK 293 cells were transfected FOXP3 together with OGT or OGA (**d**) or treated with inhibitors of OGT (ST045849) or OGA (TMG) (**e**). FOXP3 O-GlcNAcylation was determined by immunoprecipitation followed by western blotting. **f** FOXP3 stability was determined by treatment of cycloheximide (CHX) in combination with MG132 with or without OGT overexpression, DMSO was used as a control for MG132. **g**–**i** FOXP3 stability was determined in the presence of OGT/OGA overexpression (**g**), TMG (**h**), or ST045849 (**i**). Data are shown as mean ± s.e.m. *p < 0.05 by unpaired student's t-test

retrovirally transduced FOXP3 and FOXP3-5A into CD4+CD25− conventional T cells and flow cytometric analyses of transduced cells showed that the protein expression level of FOXP3-5A was much lower than that of wildtype FOXP3 (Fig. 2g, h). These results indicate that O-GlcNAcylation is required to stabilize FOXP3.

**OGT-deficiency in Treg cells leads to a scurfy phenotype**. To directly examine whether OGT regulates mature Treg cell function in vivo, we generated mice with Treg cell-specific deletion of OGT by using Cre recombinase driven by the endogenous *Foxp3* locus (*Foxp3^{YFP-Cre}*) to delete the *loxP*-flanked *Ogt* gene after FOXP3 was expressed in Treg cells. Of note, the *Foxp3* and *Ogt* genes are located about 40 centimorgans apart on the X Chromosome; thus, we were able to successfully obtain KO mice. Protein O-GlcNAcylation was specifically diminished in Treg cells but not non-Treg CD4+ T, CD8+ T, B, or natural killer cells (Supplementary Fig. 3A). Compared to *Foxp3^{YFP-Cre/Y}Ogt^{wt/Y}* control mice, *Foxp3^{YFP-Cre/Y}Ogt^{fl/Y}* male KO mice progressively developed systemic autoimmune lesions including conjunctivitis, dermatitis, hunched posture (Fig. 3a), extensive lymphadenopathy and splenomegaly (Fig. 3b), and loss of body weight (Fig. 3c). KO male mice became moribund at approximately 4 weeks of age (Fig. 3d), and massive lymphocytic infiltration could be seen in colon epithelium, skin epidermis, liver sinusoids, and lung interstitium (Fig. 3e).

We then analyzed the lymphocyte compartment at the age of 2 weeks, before an autoimmune phenotype was overtly apparent.

The absolute numbers of CD4+ and CD8+ T cells were increased in the lymph nodes (LNs) of *Foxp3^{YFP-Cre/Y} Ogt^{fl/Y}* mice (Supplementary Fig. 3B). The percentage of effector/memory cells (CD44^{hi}CD62L^{lo}) within the CD4+ and CD8+ compartments were consistently higher in both the LNs and the spleen of *Foxp3^{YFP-Cre/Y} Ogt^{fl/Y}* KO mice than those in *Foxp3^{YFP-Cre/Y} Ogt^{wt/Y}* controls (Fig. 3f, g). B cell frequency in the LNs and levels of IgG, IgM, and free kappa and lambda chains in the serum were upregulated in KO mice (Supplementary Fig. 3C, D). Moreover, the frequencies and absolute numbers of CD4+ subsets including T helper (Th) 1, Th2, and Th17 were all increased (Fig. 3h-k and Supplementary Fig. 3E-G). However, we could only observe significantly increased expression of interferon-γ (IFNγ) in CD4+FOXP3− T cells in *Foxp3^{YFP−Cre/Y} Ogt^{fl/Y}* KO mice (Fig. 3l). Expression of IL-4, IL-5, IL-13, and IL-17 remained unchanged (Fig. 3l and Supplementary Fig. 3H, I). These observations reveal an excessive Th1-dominant inflammatory response in *Foxp3^{YFP−Cre/Y} Ogt^{fl/Y}* mice.

**O-GlcNAcylation stabilizes the Treg cell lineage**. We then sought to characterize Treg cells directly. 2-week-old mice showed no difference in the Treg cell frequency among CD4+ TCRβ+ cells in the LNs or spleen (Fig. 4a, b). The absolute number of Treg cells in the LNs of *Foxp3^{YFP-Cre/Y}Ogt^{fl/Y}* mice even surpassed those in *Foxp3^{YFP-Cre/Y}Ogt^{wt/Y}* controls (Supplementary Fig. 4A, B). In the thymus, we did not observe any reduction in the frequency or number of Treg cells (Supplementary Fig. 4C, D). In addition, there was no difference in Treg

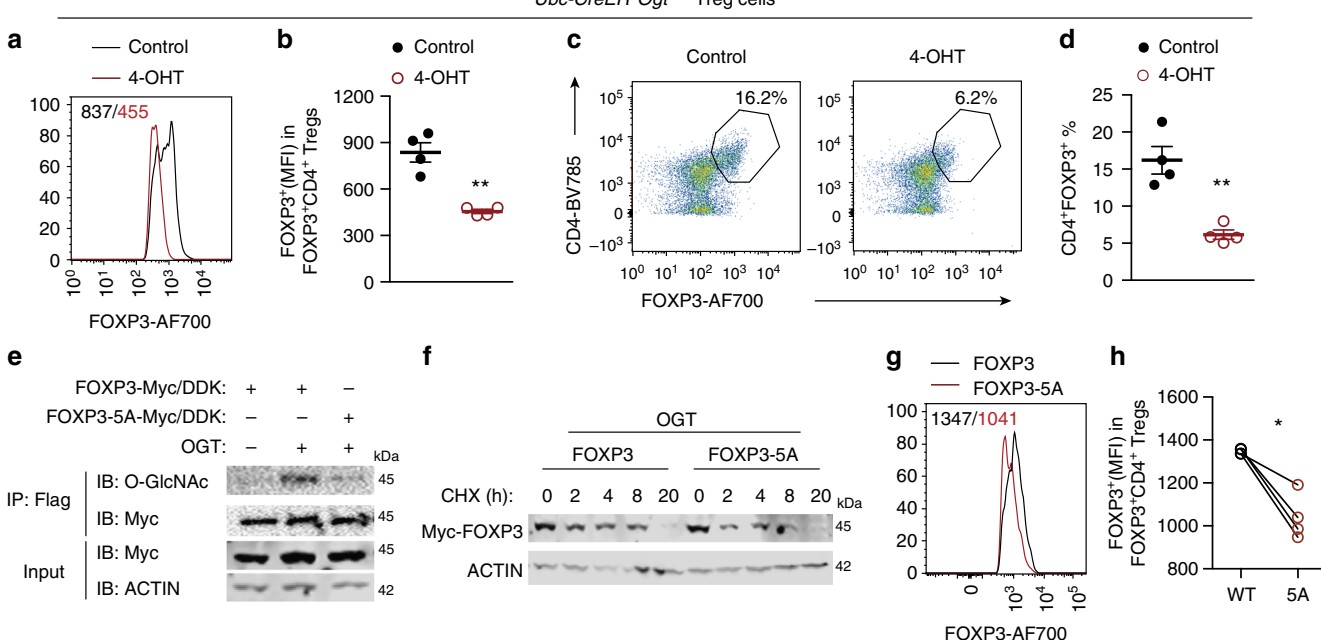

**Fig. 2** O-GlcNAcylation stabilizes the FOXP3 protein in Treg cells. **a–d** Treg cells isolated from *Ubc-CreER⁺Ogt^fl/Y* mice were treated with 4-OHT for 3‾day ex vivo, ethanol was used as the control, $n = 4$ each group. MFI of FOXP3 in CD4⁺FOXP3⁺ Treg cells was analyzed in **a** and quantified in **b**. Representative flow cytometry of CD4⁺ FOXP3⁺Treg cells was plotted in **c** and the frequencies of Treg cells were shown in **d**. **e** FOXP3 or FOXP3-5A were transfected in HEK 293 cells with or without OGT. FOXP3 O-GlcNAcylation was determined by immunoprecipitation followed by Western blotting. **f** FOXP3 and FOXP3-5A stability was determined by treatment of CHX in the presence of OGT overexpression. **g, h** CD4⁺CD25⁻ naïve T cells isolated from wildtype mice were infected with retroviruses expressing FOXP3 or FOXP3-5A in the presence of anti-CD3/CD28 beads ($n = 4$). MFI of FOXP3 was analyzed (**g**) and quantified (**h**) in CD4⁺FOXP3⁺ Treg cells. Data are shown as mean ± s.e.m. *$p < 0.05$; **$p < 0.01$ by unpaired student's $t$-test (**b, d**) and paired student's $t$-test (**h**)

cell proliferation or apoptosis between *Foxp3^YFP-Cre/Y Ogt^fl/Y* and *Foxp3^YFP-Cre/Y Ogt^wt/Y* mice, shown by Ki-67 and Annexin V staining, respectively (Supplementary Fig. 4E, F). These data suggest that OGT is dispensable for the development of thymic Treg cells.

Similar to findings in induced *Ubc-Cre/ERT2⁺Ogt^fl/Y* cells ex vivo (see Fig. 2a–d), Treg cells from 2-week-old *Foxp3^YFP-Cre/Y Ogt^fl/Y* KO mice had less FOXP3 protein expression than those from control mice (Fig. 4c, d). Treating isolated Treg cells from *Foxp3^YFP-Cre/Y Ogt^wt/Y* and *Foxp3^YFP-Cre/Y Ogt^fl/Y* mice with CHX, we were able to find decreased FOXP3 protein stability in OGT-deficient Treg cells (Supplementary Fig. 4G). In adult *Foxp3^YFP-Cre/wt Ogt^fl/fl* female mice, in which OGT-sufficient (YFP⁻) and OGT-deficient (YFP⁺) Treg cells co-existed because of random inactivation of the X chromosome, YFP⁺ OGT-deficient Treg cells displayed similar abundance as those YFP⁺ OGT-sufficient Treg cells in *Foxp3^YFP-Cre/wt Ogt^wt/fl* females (Supplementary Fig. 4H). In *Foxp3^YFP-Cre/wt Ogt^fl/fl* females, FOXP3 expression was lower in OGT-deficient Treg cells than in OGT-sufficient Treg cells (Fig. 4e). These data demonstrate that OGT is required for FOXP3 protein stability in Treg cells.

Continuous expression of FOXP3 maintains the Treg cell identity. In *Foxp3^YFP-Cre/Y Ogt^fl/Y* KO mice, a significant proportion of YFP⁺CD25⁺ cells lost FOXP3 expression when compared to their *Foxp3^YFP-Cre/Y Ogt^wt/Y* counterparts (Fig. 4f), suggesting that OGT-deficient cells tend to become the so-called ex-Treg cells or latent Treg cells[12]. To directly trace the *Foxp3* lineage, we crossed inducible *Foxp3^eGFP-Cre-ERT2/Y Ogt^fl/Y* mice to the *Rosa26^tdTomato* Cre-reporter line, in which a *loxP*-flanked STOP cassette preventing transcription of the tdTomato protein was inserted into the *Rosa26* locus[32]. Tamoxifen-containing diet feeding induced Treg cell-specific ablation of protein O-GlcNAcylation

(Supplementary Fig. 5A) and progressive systemic inflammation in *Foxp3^eGFP-Cre-ERT2/Y Ogt^fl/Y Rosa26^tdTomato/wt* mice (Supplementary Fig. 5B-E). Gating on the CD4⁺GITR⁺tdTomato⁺ population, we found that more FOXP3⁻tdTomato⁺ ex-Treg cells emerged in *Foxp3^eGFP-Cre-ERT2/Y Ogt^fl/Y Rosa26^tdTomato/wt* KO mice than in *Foxp3^eGFP-Cre-ERT2/Y Ogt^wt/Y Rosa26^tdTomato/wt* controls (Fig. 4g). These ex-Treg cells were prone to express the Th1 transcription factor T-BET and the Th2 transcription factor GATA3, but not the Th17 transcription factor RORγT (Fig. 4h, i). These results show that OGT-mediated protein O-GlcNAcylation contributes to the maintenance of Treg lineage stability.

**O-GlcNAc is indispensable for the effector Treg development**. In mature Treg cells, IL-2 and TCR signaling are not only required for lineage stability, but also for suppressive function, partially via the AKT-mTOR axis[33]. The intensity of ribosomal protein S6 phosphorylation but not AKT S473 phosphorylation was decreased in OGT-deficient Treg cells (Supplementary Fig. 6A), suggesting a mTORC1-specific defect. An in vitro assay of Treg's ability to abrogate lymphocyte proliferation did not reveal any difference between OGT-sufficient and OGT-deficient cells (Supplementary Fig. 6B), indicating that OGT is only indispensable for part of the immune functions regulated by Treg cells and/or compensatory mechanisms may be evolved. Nonetheless, in *Foxp3^YFP-Cre/Y Ogt^fl/Y* mice, we found that the abundance of CD44^hiCD62L^lo effector Treg (eTreg) cells was significantly lower than *Foxp3^YFP-Cre/Y Ogt^wt/Y* counterparts (Fig. 5a). Strikingly, eTreg cells that express signature effector molecules such as KLRG1[34], PD-1[35], and CD73[36] were almost eliminated in *Foxp3^YFP-Cre/Y Ogt^fl/Y* mice (Fig. 5b-d).

*Foxp3^YFP-Cre/wt Ogt^fl/fl* female mice were devoid of autoimmune defects (Supplementary Fig. 6C, D), since they possessed

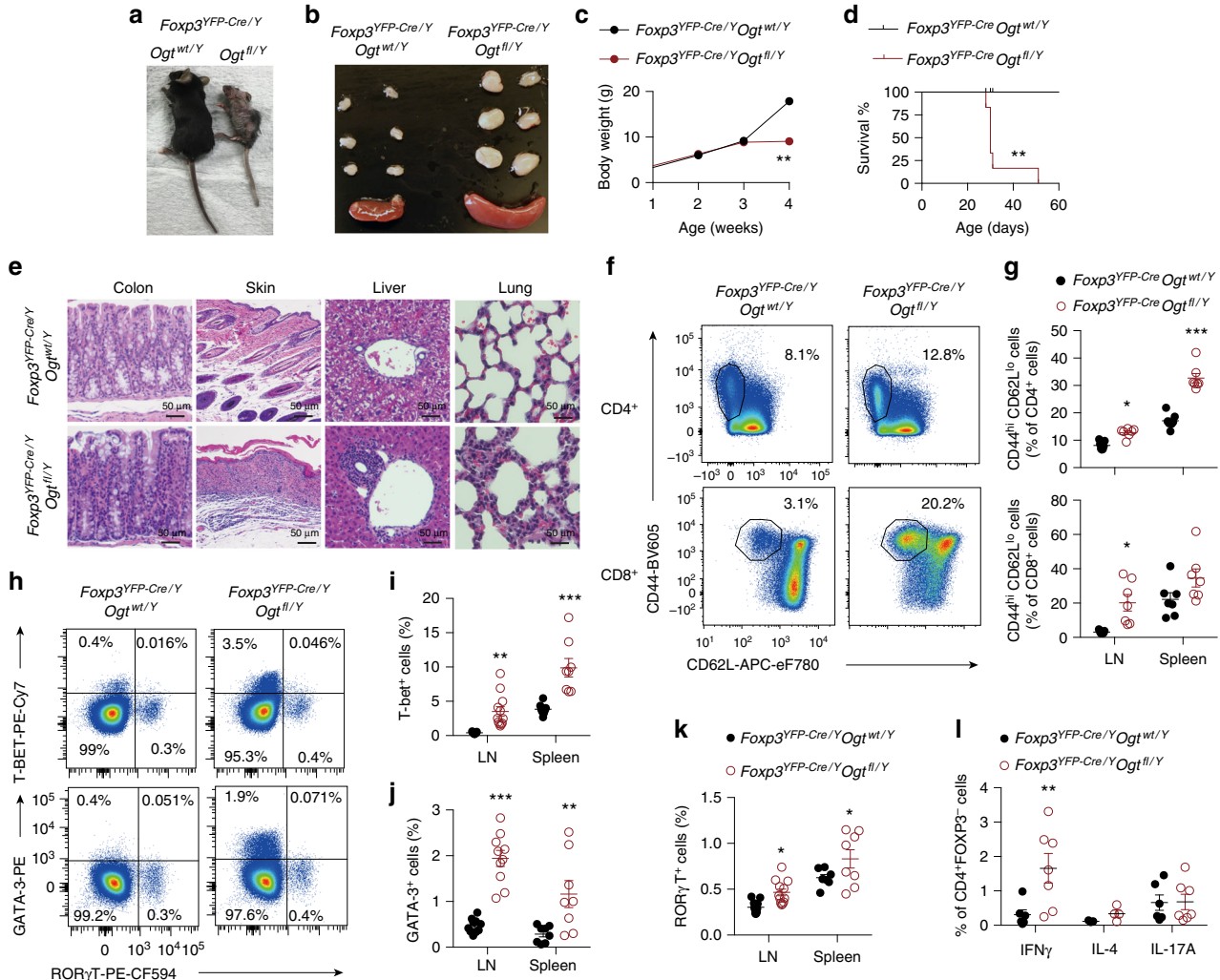

**Fig. 3** OGT-deficiency in Treg cells leads to a scurfy phenotype in male mice. **a**, **b** Representative images of 4-week-old $Foxp3^{YFP-Cre/Y}$ $Ogt^{wt/Y}$ and $Foxp3^{YFP-Cre/Y}$ $Ogt^{fl/Y}$ mice (**a**) and peripheral LNs and spleen (**b**). **c**, **d** Body weight curve (**c**, $n = 5$) and survival curve (**d**, $n = 6$) of $Foxp3^{YFP-Cre/Y}$ $Ogt^{wt/Y}$ and $Foxp3^{YFP-Cre/Y}$ $Ogt^{fl/Y}$ mice. **e** Representative images of H&E staining of the colon, skin, liver, and lung from 4-week-old $Foxp3^{YFP-Cre/Y}$ $Ogt^{wt/Y}$ and $Foxp3^{YFP-Cre/Y}$ $Ogt^{fl/Y}$ mice. **f**, **g** Representative flow cytometry plots showing CD44$^{hi}$CD62L$^{lo}$ effector T cells among CD4$^+$ (top) and CD8$^+$ (bottom) T cells in the LNs (**f**) and quantification of the frequencies in the LNs and spleen of 2-week-old $Foxp3^{YFP-Cre/Y}$ $Ogt^{wt/Y}$ and $Foxp3^{YFP-Cre/Y}$ $Ogt^{fl/Y}$ mice, at least $n = 6$ each group (**g**). **h–k** Representative flow cytometry plots showing T-BET$^+$, GATA3$^+$ and ROR$\gamma$T$^+$ cells among CD4$^+$ T cells in the LNs (**h**) and quantification of the frequencies of T-BET$^+$ cells (**i**), GATA3$^+$ cells (**j**) and ROR$\gamma$T$^+$ cells (**k**) in the LNs and spleen of 2-week-old $Foxp3^{YFP-Cre/Y}$ $Ogt^{wt/Y}$ and $Foxp3^{YFP-Cre/Y}$ $Ogt^{fl/Y}$ mice were shown, $n = 8$ each group. **l** Frequencies of IFN$\gamma^+$, IL-4$^+$ and IL-17A$^+$ cells in CD4$^+$FOXP3$^-$ T cells stimulated with PMA/Ionomycin in the LNs of 2-week-old $Foxp3^{YFP-Cre/Y}$ $Ogt^{wt/Y}$ and $Foxp3^{YFP-Cre/Y}$ $Ogt^{fl/Y}$ mice, at least $n = 3$ each group. Data are shown as mean ± s.e.m. *$p < 0.05$; **$p < 0.01$; ***$p < 0.001$ by unpaired student's $t$-test (**c**, **g**, and **i–l**) and Kaplan-Meier Analysis (**d**)

both OGT-sufficient (YFP$^-$) and OGT-deficient (YFP$^+$) Treg cells because of random inactivation of the X chromosome. Nevertheless, we could observe significant reductions in the abundance of CD44$^{hi}$CD62L$^{lo}$, KLRG1$^+$, PD-1$^+$, CD73$^+$, and CD103$^+$ eTreg cells when comparing YFP$^+$ OGT-deficient to YFP$^-$ OGT-sufficient cells in the same $Foxp3^{YFP-Cre/wt}Ogt^{fl/fl}$ mice (Fig. 5e-i). Similar reductions in eTreg signature molecules could be observed when comparing YFP$^+$ Treg cells between $Foxp3^{YFP-Cre/wt}Ogt^{fl/fl}$ and $Foxp3^{YFP-Cre/wt}Ogt^{wt/fl}$ mice (Supplementary Fig. 6E-I). These data indicate a cell-intrinsic loss of effector molecules in OGT-deficient Treg cells.

**Attenuated IL-2/STAT5 signaling in OGT-deficient Treg cells.** It seemed unlikely to us that the mild downregulation of FOXP3 protein stability could account for the absence of eTreg cell formation, as FOXP3 binds to the same set of regulatory elements in both resting Treg cells and eTreg cells[37]. To gain comprehensive

insight into the OGT-dependent transcriptional program in Treg cells, we performed RNA-sequencing of isolated YFP$^+$ Treg cells from $Foxp3^{YFP-Cre/wt}Ogt^{wt/fl}$ and healthy $Foxp3^{YFP-Cre/wt}Ogt^{fl/fl}$ females to avoid secondary changes in gene expression caused by inflammation. We were able to identify 269 differentially expressed genes including 154 downregulated and 115 upregulated with p values less than 0.01 (Fig. 6a). eTreg cell markers such as $klrg1$, $S100a4$, $Gzmb$, and $Ccr2$ were downregulated by the loss of O-GlcNAcylation (Fig. 6a). The transcription factor B lymphocyte-induced maturation protein (BLIMP)-1 is common to all eTreg cells[38], and we found that BLIMP-1-upregulated genes were enriched in OGT-sufficient Treg cells, while BLIMP-1-downregulated genes were enriched in OGT-deficient Treg cells (Fig. 6b, c), suggesting that OGT maintains a transcriptional program similar to that of BLIMP-1$^+$ eTreg cells. Ingenuity Pathway Analysis (IPA) identified 9 potential upstream regulators which were all predicted to be inhibited, and the top one was IL-2

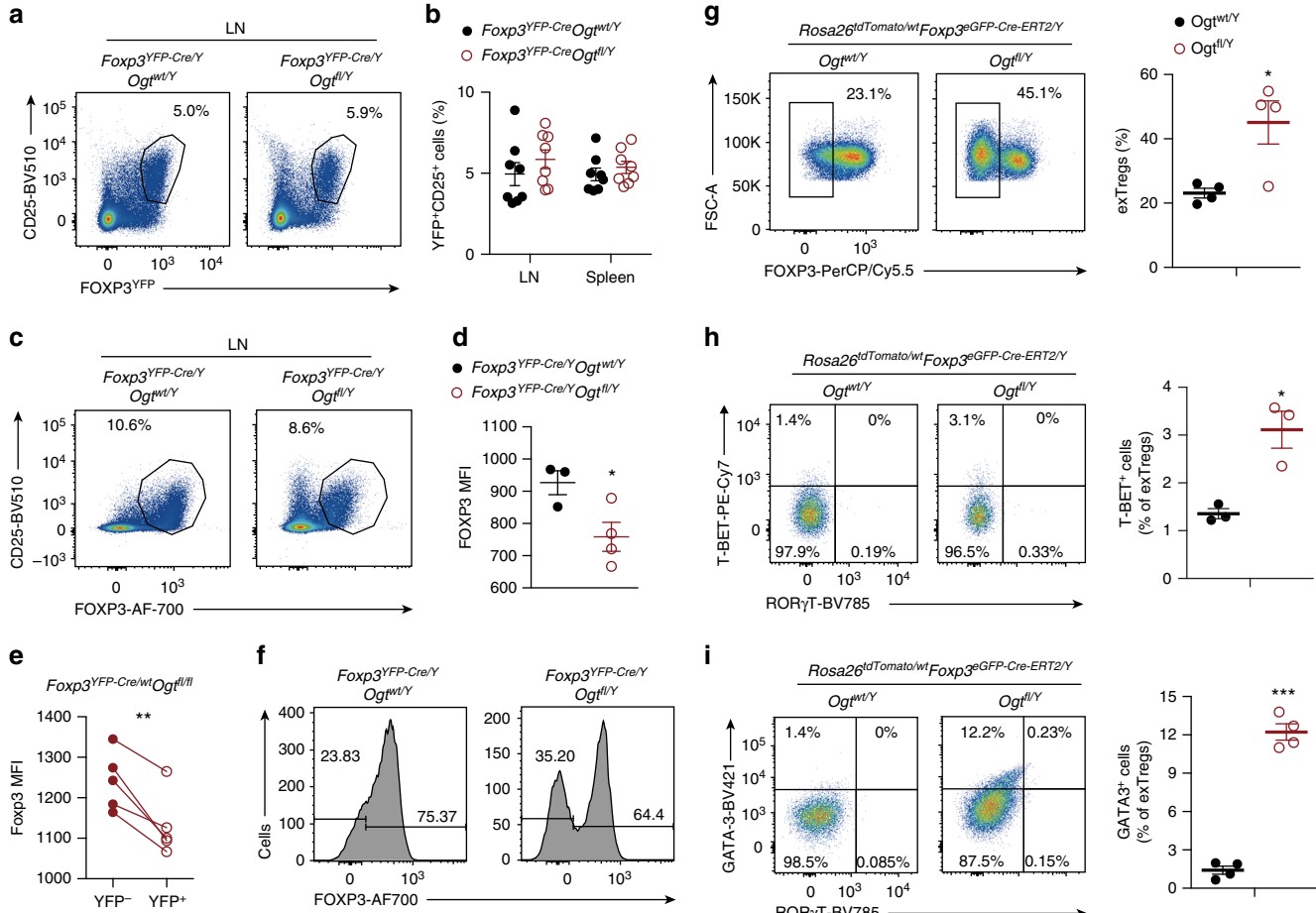

**Fig. 4** Protein O-GlcNAcylation stabilizes the Treg cell lineage. **a**, **b** Flow cytometry of YFP+CD25+ cells among CD4+ T cells in the LNs (**a**) and frequencies of YFP+CD25+ cells in the LNs and spleen from 2-week-old $Foxp3^{YFP-Cre/Y} Ogt^{wt/Y}$ and $Foxp3^{YFP-Cre/Y} Ogt^{fl/Y}$ mice, $n = 8$ each group (**b**). **c**, **d** Flow cytometry of FOXP3+CD25+ cells among CD4+ T cells (**c**) and MFI of FOXP3 in FOXP3+CD25+ Treg cells (**d**, $n = 3$–4) in the LNs from 2-week-old $Foxp3^{YFP-Cre/Y} Ogt^{wt/Y}$ and $Foxp3^{YFP-Cre/Y} Ogt^{fl/Y}$ mice. **e** MFI of FOXP3 in OGT-sufficient and -deficient Treg cells in the LNs from $Foxp3^{YFP-Cre/wt} Ogt^{fl/fl}$ mice, $n = 5$. **f** Histogram of FOXP3 expression in CD4+CD25+YFP+ Treg cells in the LNs from 2-week-old $Foxp3^{YFP-Cre/Y} Ogt^{wt/Y}$ and $Foxp3^{YFP-Cre/Y} Ogt^{fl/Y}$ mice, $n = 6$ each group. **g** Flow cytometry and quantification of the frequencies of Td-tomato+GITR+FOXP3- ex-Treg cells among CD4+ T cells in the LNs from $Foxp3^{eGFP-Cre-ERT2/Y} Ogt^{wt/Y} Rosa26^{tdTomato/wt}$ and $Foxp3^{eGFP-Cre-ERT2/Y} Ogt^{fl/Y} Rosa26^{tdTomato/wt}$ mice, $n = 4$. **h**, **i** T-BET+, GATA3+ and RORγT+ cells among ex-Treg cells in the LNs from $Foxp3^{eGFP-Cre-ERT2/Y} Ogt^{wt/Y} Rosa26^{tdTomato/wt}$ and $Foxp3^{eGFP-Cre-ERT2/Y} Ogt^{fl/Y} Rosa26^{tdTomato/wt}$ mice, $n = 3$–4. Data are shown as mean ± s.e.m. *$p < 0.05$; **$p < 0.01$; ***$p < 0.001$ by unpaired (**b**, **d**, **g**–**i**) and paired (**e**) student's $t$-test

(Fig. 6d). To evaluate the responsiveness of Treg cells to IL-2, we treated mice with immune complexes consisting of mouse IL-2 and anti-IL-2 antibody to expand eTreg cells (Supplementary Fig. 7A)[34,39]. IL-2 immune complex treatment moderately enlarged the LNs and spleen but did not change total protein O-GlcNAcylation in Treg cells (Supplementary Fig. 7B, C), suggesting that IL-2 does not directly regulate O-GlcNAc signaling. However, protein O-GlcNAcylation was required for IL-2-stimulated development of eTreg cells, as the expansion of KLRG1+, GZMB+, and BLIMP-1+ eTreg populations was absent in OGT-deficient Treg cells (Fig. 6e-j and Supplementary Fig. 7D). The increase in the expression of CD25 and IL-10 induced by IL-2 in OGT-sufficient cells was also inhibited when OGT was deleted (Supplementary Fig. 7E, F).

STAT5, acting downstream of IL-2, is indispensable for the formation of KLRG1+ terminally differentiated Treg cells and their suppressive function[18,34]. O-GlcNAcylation of STAT5 has been shown to promote its oligomerization and transcriptional activity in cooperation with tyrosine phosphorylation in neoplastic cells[40]. Consistent with the defective IL-2 response, many STAT5-target genes were downregulated in OGT-deficient Treg cells (Fig. 6k). However, the IL-2-activated tyrosine phosphorylation of STAT5 (pY-STAT5) in Treg cells was not affected by the loss of OGT (Supplementary Fig. 8A, B), again supporting the notion that O-GlcNAcylation can act independently of tyrosine phosphorylation[40]. O-GlcNAcylation seemed not to regulate the cellular localization of STAT5 either (Supplementary Fig. 8C). Retroviral expression of a constitutively active STAT5A (cS5) in Treg cells increased the expression of the STAT5-target gene *Socs1* and *Socs3*, whereas the O-GlcNAc-deficient cS5-T92A diminished such effect[40] (Fig. 6l, m). Moreover, in OGT-deficient Treg cells, reconstitution of cS5 but not cS5-T92A increased *Socs1* expression (Supplementary Fig. 8D), suggesting that STAT5 O-GlcNAcylation is important for its transcriptional activity. Collectively, these data demonstrate that the deficiency in protein O-GlcNAcylation results in attenuated IL-2/STAT5 activity in Treg cells.

**STAT5B-CA partially rescues Treg cell dysfunction.** Treg cell-specific deletion of OGT resulted in a scurfy phenotype with comparable early onset and disease severity to those observed upon IL-2R or STAT5 ablation[18]. Expression of a constitutively

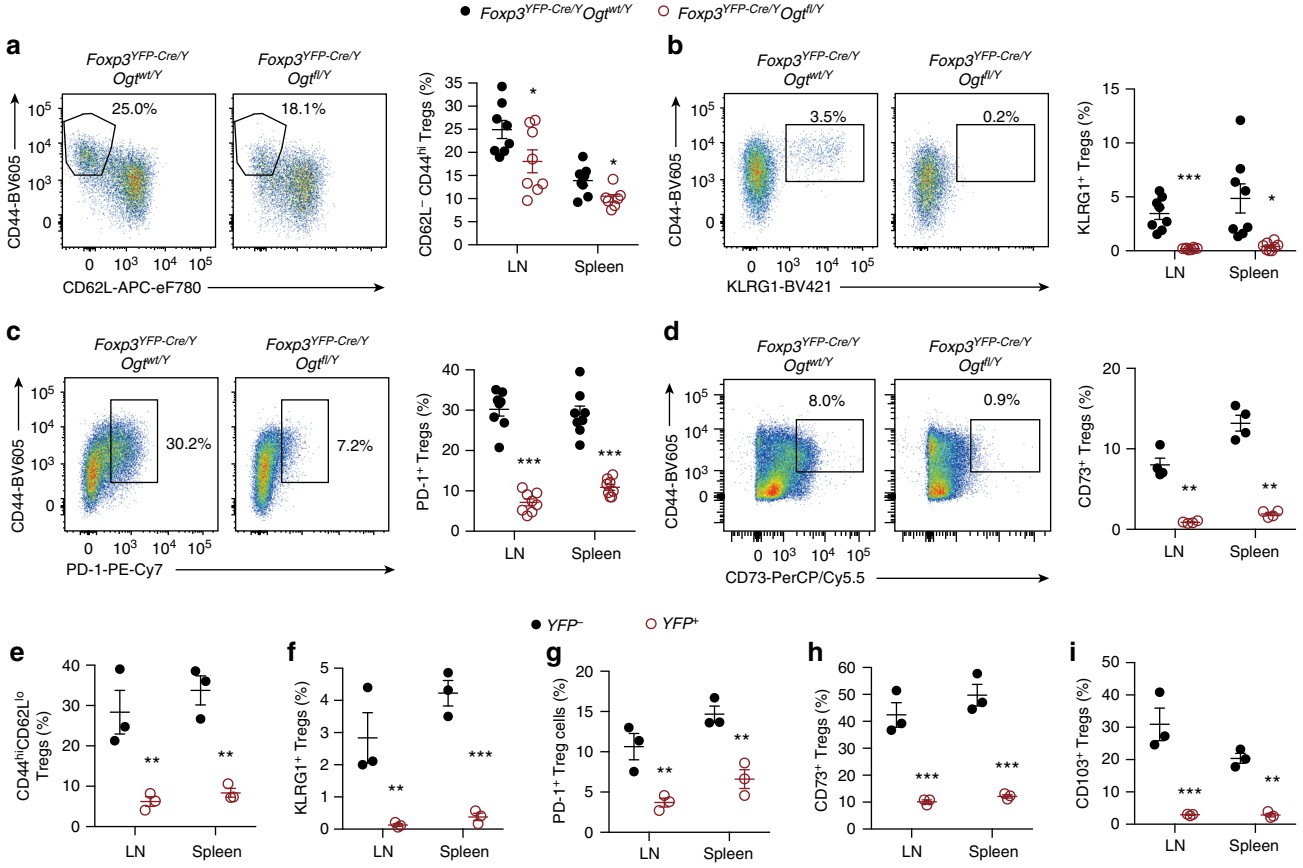

**Fig. 5** O-GlcNAcylation is required for the effector differentiation of Treg cells. **a–d** Frequencies of CD44$^{hi}$CD62L$^{lo}$ (**a**), CD44$^+$KLRG1$^+$ (**b**), CD44$^+$PD-1$^+$ (**c**), and CD44$^+$CD73$^+$ (**d**) cells among YFP$^+$CD25$^+$ Treg cells in the LNs and spleen from 2-week-old *Foxp3$^{YFP-Cre/Y}$ Ogt$^{wt/Y}$* and *Foxp3$^{YFP-Cre/Y}$ Ogt$^{fl/Y}$* mice, at least $n = 4$ each group. **e–i** Frequencies of indicated cell populations among YFP$^-$OGT-sufficient and YFP$^+$ OGT-deficient CD4$^+$TCRβ$^+$CD25$^+$ GITR$^+$ Treg cells in the LNs and spleen from *Foxp3$^{YFP-Cre/wt}$ Ogt$^{fl/fl}$* mice, $n = 3$ each group. Data are shown as mean ± s.e.m. *$p < 0.05$; **$p < 0.01$; ***$p < 0.001$ by unpaired student's *t*-test

active form of STAT5B-CA partially rescues IL-2R deficient Treg function[18]. We next sought to determine whether STAT5 activation could restore Treg cell function in *Foxp3$^{YFP-Cre/Y}$ Ogt$^{fl/Y}$* mice by crossing them to a Cre-inducible line overexpressing STAT5B-CA at the *Rosa26* locus (*Rosa26$^{Stat5b-CA}$*)[18]. Remarkably, the *Foxp3$^{YFP-Cre/Y}$Ogt$^{fl/Y}$Rosa26$^{Stat5b-CA/wt}$* mice with STAT5B-CA overexpression specifically in OGT-deficient Treg cells alleviated skin inflammation (Fig. 7a), reduced sizes of the LNs and spleen (Fig. 7b), and prolonged lifespan (Fig. 7c), when compared to *Foxp3$^{YFP-Cre/Y}$Ogt$^{fl/Y}$* mice. Consistent with the role of STAT5 in regulating *Foxp3* transcription and Treg cell development[3,18,41], FOXP3 protein expression was restored to a level comparable to wildtype mice and the frequency of Treg cells was further increased when STAT5B-CA was present (Fig. 7d, e, and Supplementary Fig. 9A). Meanwhile, the eTreg cell population was significantly boosted to a level comparable to wildtype mice, shown as CD44$^{hi}$CD62L$^{lo}$ (Fig. 7f and Supplementary Fig. 9B) and CD44$^+$KLRG1$^+$ subsets (Fig. 7g and Supplementary Fig. 9C). However, the expression levels of CD73 and PD-1 on a per-cell basis did not change (Supplementary Fig. 9D), suggesting a partial rescue in the effector function of Treg cells by STAT5B-CA. Moreover, the fraction of YFP$^+$ OGT-deficient Treg cells that lost FOXP3 expression as a result of protein instability was substantially inhibited by the presence of STAT5B-CA (Fig. 7h and Supplementary Fig. 9E), indicating that STAT5B-CA also restored Treg lineage stability.

Accordingly, STAT5B-CA overexpression reduced the number of total CD4$^+$ and CD8$^+$ T cells (Fig. 7i, j) and the frequency of

memory/effector CD4$^+$ and CD8$^+$ T cells in *Foxp3$^{YFP-Cre/Y}$Ogt$^{fl/Y}$ Rosa26$^{Stat5b-CA/wt}$* mice (Fig. 7k, l, and Supplementary Fig. 9F). In addition, STAT5B-CA overexpression prevented the increase in Th1 and Th2 cells caused by loss of O-GlcNAcylation in Treg cells (Fig. 7m-o and Supplementary Fig. 9G) and reduced the production of IFNγ by CD4$^+$ and CD8$^+$ T cells (Fig. 7p and Supplementary Fig. 9H, I). Collectively, these data show that restoring STAT5 signaling in OGT-deficient Treg cells improves lineage stability and effector function, thereby delaying the autoimmune responses in mice.

**O-GlcNAc promotes the suppressive program of Tregs**. The therapeutic potential of Treg cells for autoimmune disorders and graft-versus-host disease (GVHD) has been well characterized in preclinical and early clinical studies[42,43]. We then asked whether enhancing protein O-GlcNAcylation could augment the suppressive function of Treg cells. We first treated mouse Treg cells with TMG to inhibit OGA (Fig. 8a), the enzyme removing O-GlcNAc moieties from proteins[44], and found that elevating protein O-GlcNAcylation by TMG increased the level of FOXP3 protein on a per cell basis (Fig. 8b, c) and the number of CD4$^+$FOXP3$^+$ Treg cells (Supplementary Fig. 10A), again demonstrating that O-GlcNAcylation stabilizes FOXP3 and Treg cell lineage. In addition, TMG treatment enhanced the expression of STAT5-target genes *Socs1* and *Socs3* (Fig. 8d, e), and the expression of activation markers including *Klrg1*, *Pd-1* and, to a lesser extent, *Cd73* and *Cd44* in mouse Treg cells (Fig. 8f).

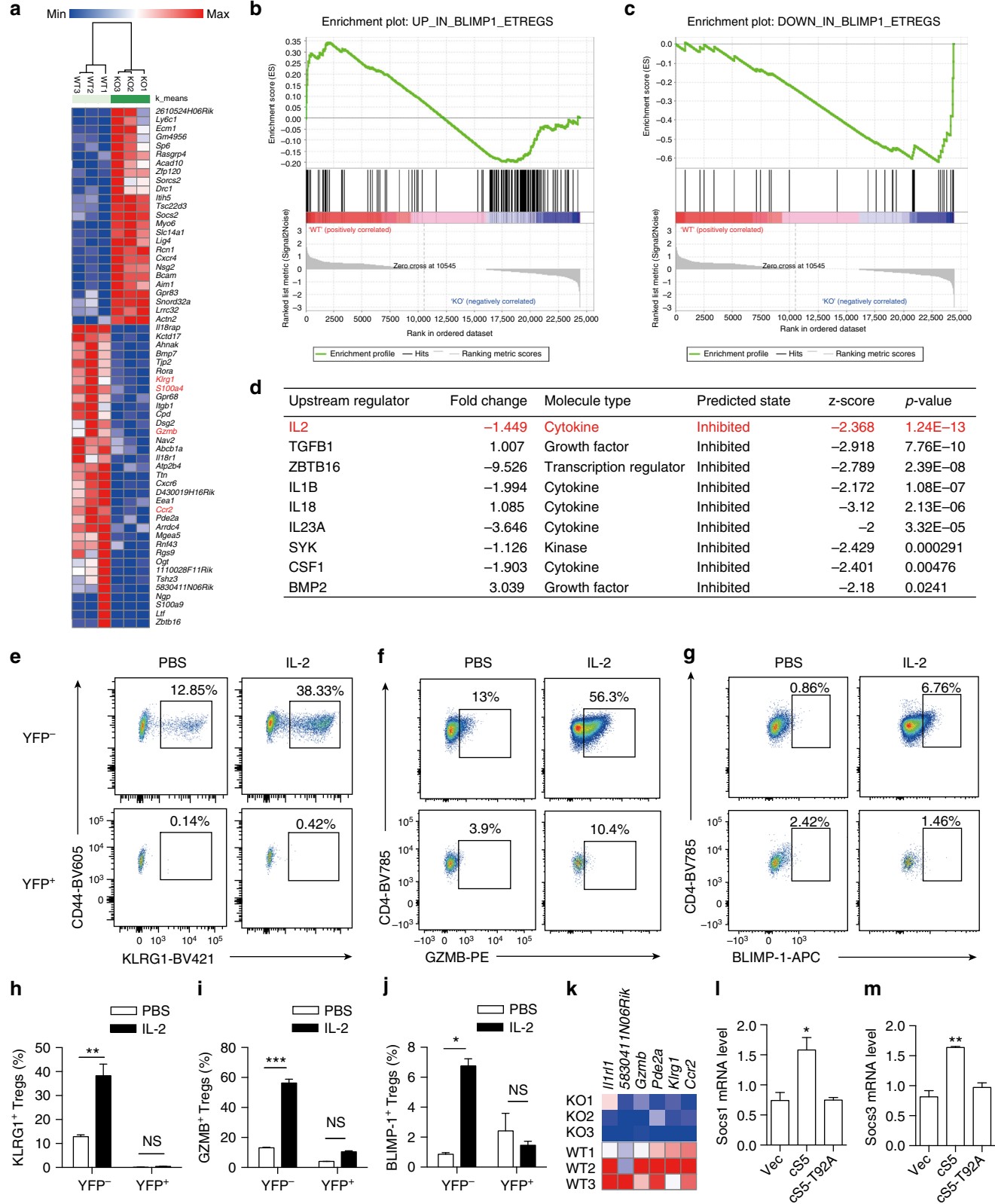

8    NATURE COMMUNICATIONS | (2019)10:354 | https://doi.org/10.1038/s41467-019-08300-3 | www.nature.com/naturecommunications

Finally, we purified CD4$^+$CD25$^+$CD127$^-$CD45RA$^+$ naïve human thymus-derived Treg (tTreg) cells from peripheral blood mononuclear cells (PBMNCs)[45]. OGA inhibition by TMG significantly increased intracellular O-GlcNAc levels (Fig. 8g) but had no effect on FOXP3 expression or tTreg cell expansion (Fig. 8h and Supplementary Fig. 10B), indicating that the endogenous O-GlcNAcylation was sufficient to maintain human FOXP3 stability. Nevertheless, we observed a significant increase in the expression of SOCS3 gene (Fig. 8i) and the suppressive activity of human Treg cells when treated with TMG (Fig. 8j, k,

**Fig. 6** Attenuated IL-2/STAT5 signaling in OGT-deficient Treg cells. **a** Hierarchical clustering of top differentially expressed genes (60 genes with FDR q values less than 0.1) between OGT-sufficient and OGT-deficient Treg cells using Morpheus. **b**, **c** Enrichment plots for up-regulated (**b**) and down-regulated (**c**) genes in Blimp1[+] eTreg cells from the Gene Set Enrichment Analysis (GSEA) of differentially expressed genes between OGT-sufficient and OGT-deficient Treg cells. **d** Ingenuity Pathway Analysis (IPA) of predicted upstream regulators for observed changes in gene expression. **e–j** $Foxp3^{YFP-Cre/wt}$ $Ogt^{fl/fl}$ female mice were injected with PBS or the IL-2 immune complex ($n = 2–5$) for 3 consecutive days and Treg cells were analyzed at day 6. Flow Cytometry of CD44[+]KLRG1[+](**e**), CD4[+]GZMB[+] (**f**) and CD4[+]BLIMP-1[+] (**g**) Treg cells among YFP[−]OGT-sufficient and YFP[+] OGT-deficient Treg cells (CD4[+]CD25[+]GITR[+]) in LNs and spleen and corresponding frequencies in **h–j**. **k** Expression of STAT5-target genes in OGT-sufficient and OGT-deficient Treg cells. **l–m** mRNA levels of $Socs1$ (**l**) and $Socs3$ (**m**) in OGT-sufficient Treg cells after retrovirus infection as indicated, $n = 3$ each group. Data are shown as mean ± s.e.m. *$p < 0.05$; **$p < 0.01$; ***$p < 0.001$ by two-way ANOVA (**h–j**) and one-way ANOVA (**l**, **m**)

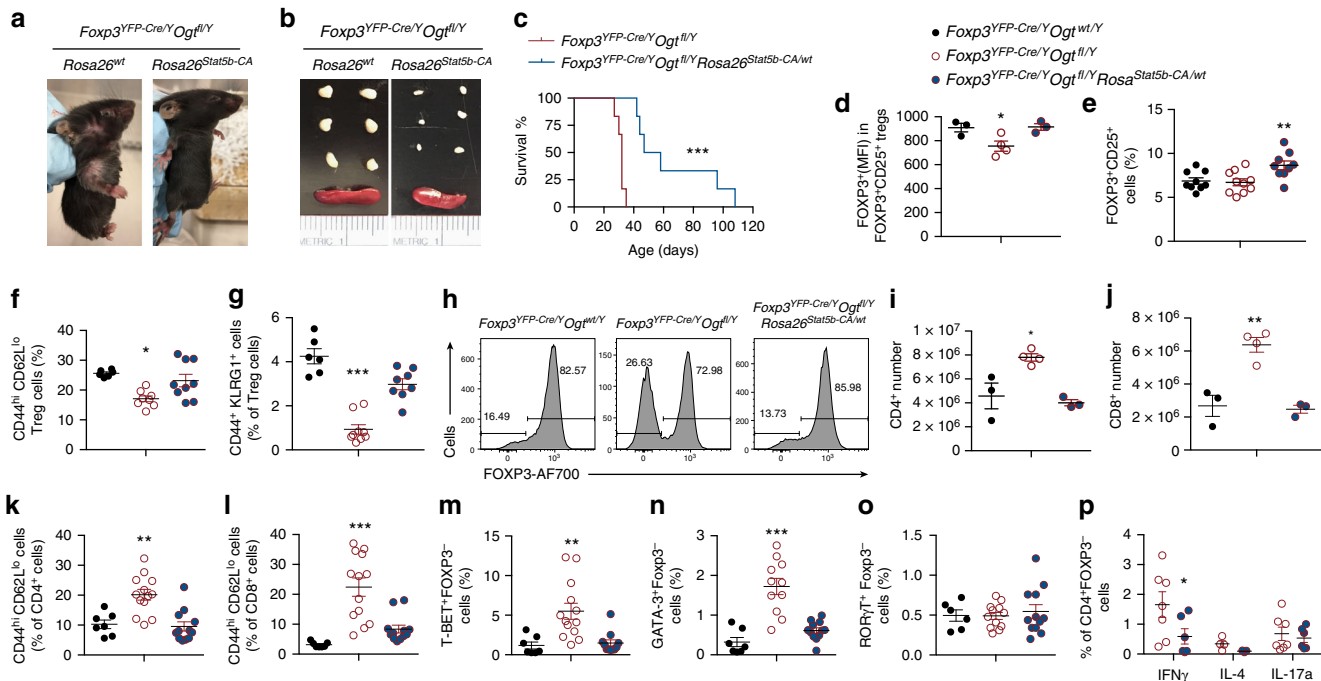

**Fig. 7** Constitutive activation of STAT5 partially rescues Treg cell dysfunction. **a**, **b** Representative images of mice (**a**) and peripheral LNs and spleen (**b**) from 2-week-old $Foxp3^{YFP-Cre/Y}$ $Ogt^{fl/Y}$ and $Foxp3^{YFP-Cre/Y}Ogt^{fl/Y}$ $Rosa26^{Stat5b-CA/wt}$ mice. **c** Survival curve of $Foxp3^{YFP-Cre/Y}$ $Ogt^{fl/Y}$ and $Foxp3^{YFP-Cre/Y}Ogt^{fl/Y}$ $Rosa26^{Stat5b-CA/wt}$ mice ($n = 6$). **d** MFI of FOXP3 in FOXP3[+]CD25[+] Treg cells in the LNs from 2-week-old $Foxp3^{YFP-Cre/Y}$ $Ogt^{wt/Y}$, $Foxp3^{YFP-Cre/Y}$ $Ogt^{fl/Y}$ and $Foxp3^{YFP-Cre/Y}Ogt^{fl/Y}$ $Rosa26^{Stat5b-CA/wt}$ mice, at least $n = 3$ each group. **e** Frequencies of FOXP3[+]CD25[+] Treg cells among CD4[+] T cells, at least $n = 9$ each group. **f–g** Frequencies of CD44[hi]CD62L[lo] (**f**) and CD44[+]KLRG1[+] (**g**) eTreg cells among FOXP3[+]CD25[+] Treg cells in the LNs from 2-week-old $Foxp3^{YFP-Cre/Y}$ $Ogt^{wt/Y}$, $Foxp3^{YFP-Cre/Y}$ $Ogt^{fl/Y}$ and $Foxp3^{YFP-Cre/Y}Ogt^{fl/Y}Rosa26^{Stat5b-CA/wt}$ mice, at least $n = 6$ each group. **h** Histogram of FOXP3 expression in CD4[+]CD25[+]YFP[+] Treg cells in the LNs from 2-week-old $Foxp3^{YFP-Cre/Y}$ $Ogt^{wt/Y}$, $Foxp3^{YFP-Cre/Y}$ $Ogt^{fl/Y}$ and $Foxp3^{YFP-Cre/Y}Ogt^{fl/Y}Rosa26^{Stat5b-CA/wt}$ mice. **i**, **j** Absolute number of CD4[+] (**i**) and CD8[+] (**j**) T cells in the LNs, at least n = 3 each group. **k**, **l** Frequencies of CD44[hi]CD62L[lo] effector T cells in CD4[+] (**k**) and CD8[+] cells (**l**), at least $n = 7$ each group. **m–o** Frequencies of T-BET[+] (**m**), GATA3[+] (**n**) and RORγT[+] (**o**) populations among CD4[+]FOXP3[−] cells in the LNs, at least $n = 6$ each group. **p** Frequencies of INFγ, IL-4 and IL-17A-producing CD4[+]FOXP3[−] cells from $Foxp3^{YFP-Cre/Y}$ $Ogt^{fl/Y}$ and $Foxp3^{YFP-Cre/Y}Ogt^{fl/Y}$ $Rosa26^{Stat5b-CA/wt}$ mice, at least $n = 3$ each group. Data are shown as mean ± s.e.m. *$p < 0.05$; **$p < 0.01$; ***$p < 0.001$ by Kaplan-Meier Analysis (**c**), unpaired student's $t$-test (**p**) and one-way ANOVA (**d–g**, **i–o**)

and Supplementary Fig. 10C). Collectively, these results reveal a fundamental role for O-GlcNAc signaling in supporting the suppressive program of mouse and human Treg cells.

## Discussion

In mature Treg cells, FOXP3 expression and signals from TCR and IL-2R are continuously required for Treg lineage stability and suppressive function[9,10,16–18,46]. In this study, we have demonstrated that, in response to TCR activation, protein O-GlcNAcylation stabilizes FOXP3 and activates STAT5, thus integrating critical signaling nodes for Treg cell homeostasis and function.

Our data suggest that O-GlcNAcylation promotes FOXP3 stability by counteracting with ubiquitination. Prior studies have

demonstrated that polyubiquitination marks FOXP3 for protein degradation via the proteasome[47–49]. Treatment of Treg cells with a pan-deubiquitinase inhibitor reduces FOXP3 protein levels and decreases suppressive activity[31]. Ubiquitination dynamics of FOXP3 is dictated by the ubiquitin ligase STUB1 and the deubiquitinase USP7[30,31]. We postulate that O-GlcNAcylation inhibits STUB1-mediated polyubiquitination and/or facilitates USP7-mediated deubiquitination to stabilize FOXP3. By identifying and mutating O-GlcNAc sites on the FOXP3 protein, we were able to show reduced FOXP3 stability when O-GlcNAcylation was ablated (Fig. 2), suggesting a potential direct interplay between O-GlcNAcylation and ubiquitination. Nonetheless, O-GlcNAc signaling can modulate proteasome activity and other posttranslational modifications to indirectly control FOXP3 protein abundance[26].

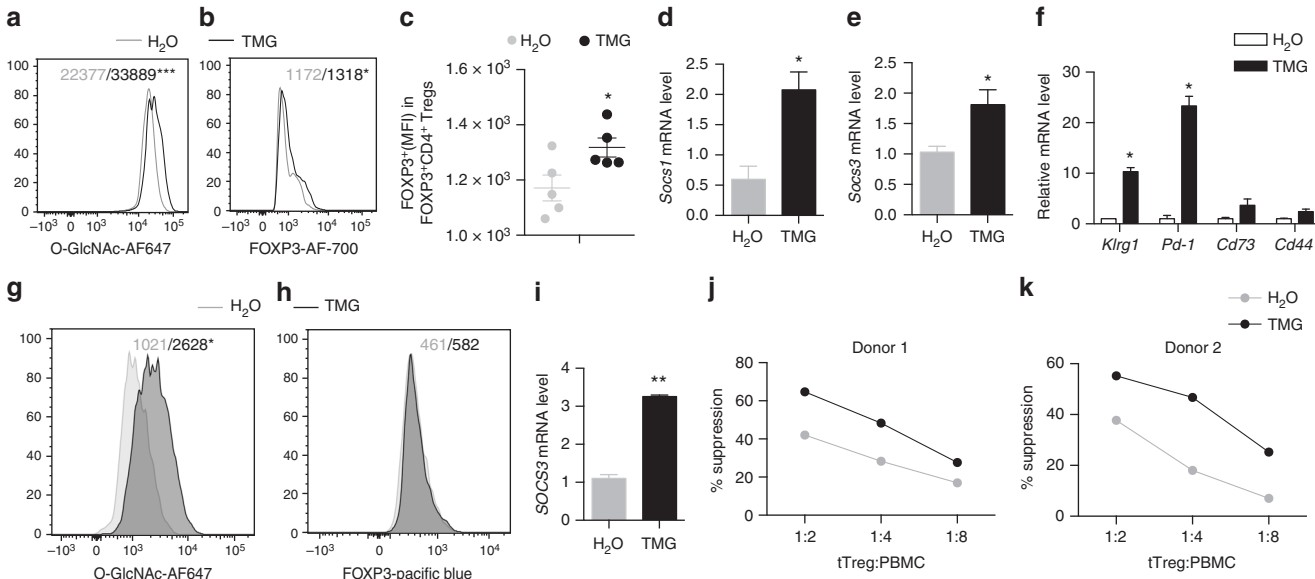

**Fig. 8** Activating O-GlcNAcylation promotes Treg cell suppressive function. **a–c** MFI of protein O-GlcNAcylation (**a**), FOXP3 (**b**, **c**) in mouse CD4$^+$FOXP3$^+$ Treg cells treated with TMG, H$_2$O treatment was used as a control ($n$ = 5). **d**, **e** mRNA levels of *Socs1* (**d**) and *Socs3* (**e**) in mouse Treg cells treated with TMG, H$_2$O treatment was used as a control ($n$ = 3). **f** Gene expression in iTreg cells treated with TMG, H$_2$O treatment was used as a control ($n$ = 2). **g**, **h** MFI of protein O-GlcNAcylation (**g**) and FOXP3 (**h**) in human tTreg cells treated with TMG for 7 days ($n$ = 3), H$_2$O treatment was used as a control. **i** mRNA levels of SOCS3 in human tTreg cells treated with TMG, H$_2$O treatment was used as a control ($n$ = 3). **j**, **k** Suppression activity of human tTreg cells treated with TMG for 7 days, H$_2$O treatment was used as a control. Data are shown as mean ± s.e.m. *$p$ < 0.05 by unpaired student's $t$-test

*Foxp3$^{YFP\text{-}Cre/Y}$Ogt$^{fl/Y}$* mice developed a scurfy phenotype at a similar pace as observed in mice with germline *Foxp3* deficiency[6–8]. When OGT was absent, FOXP3 stability was only modestly reduced ex vivo and in vivo (Figs. 2 and 4). Thus, FOXP3 instability is only one of the contributors to Treg cell dysfunction observed in *Foxp3$^{YFP\text{-}Cre/Y}$Ogt$^{fl/Y}$* mice. In Treg cells with TCR ablation, there was also a slight reduction in FOXP3 protein expression, without affecting *Foxp3* transcription[16,17]. It would be interesting to determine whether TCR signaling controls FOXP3 stability through protein O-GlcNAcylation.

In conventional T (Tconv) cells, O-GlcNAcylated proteins are abundant and dynamically regulated by TCR activation[50–52]. These proteins are involved in many biological processes, including TCR signaling (LCK, ZAP70, STAT3, c-JUN, NFAT, etc), transcriptional and translational processes, and mRNA processing. Compared to Tconv cells, Treg cells possess a unique proteomic signature to accommodate specific metabolic requirements and regulatory function[53,54]. It is unclear if O-GlcNAcylated proteins identified in Tconv cells can be modified in Treg cells. Future O-GlcNAc proteomics is needed to not only profile O-GlcNAcylated proteins in response to TCR activation but also identify other potential targets that could regulate Treg cell function. Nevertheless, it is unlikely that TCR signaling is a major downstream pathway that mediates the effect of O-GlcNAcylation, as the activation of immune system in Treg-specific TCR deletion mice was much milder than we observed in *Foxp3$^{YFP\text{-}Cre/Y}$Ogt$^{fl/Y}$* mice[16,17].

Through gene expression profiling, we predicted that the IL-2/STAT5 signaling pathway was inhibited in OGT-deficient Treg cells. In accordance with the prediction, we found that eTreg cells failed to expand in response to IL-2 immune complexes when OGT was ablated and the scurfy phenotype in *Foxp3$^{YFP\text{-}Cre/Y}$Ogt$^{fl/Y}$* mice could be partially ameliorated by a constitutive-active form of STAT5B. In OGT-deficient Treg cells, even though the expression of STAT5 target genes was significantly downregulated, we did not observe any change in levels of pY-STAT5, suggesting that O-GlcNAcylation can

function independently of tyrosine phosphorylation. This is consistent with a recent study showing that O-GlcNAcylation and tyrosine phosphorylation act together to activate STAT5-depedent malignancies[40]. Although O-GlcNAcylation at threo-nine 92 is required for strong pY-STAT5A upon IL-3 stimulation, erythropoietin-activated pY-STAT5A and IL-2-stimulated T cell proliferation were comparable between cells expressing constitutively active STAT5A (cS5) and cS5-T92A that lacks O-GlcNAcylation. In addition, there was no significant change in pY-STAT5B when introducing the T92A mutant to human STAT5B or the hyperactive N642H derivative[40]. These findings indicate that O-GlcNAcylation and IL-2-induced pY-STAT5 have separate but additive effects on STAT5 function.

In TCR-triggered naïve T cells, glucose and glutamine flux into the hexosamine biosynthetic pathway (HBP) to generate UDP-GlcNAc, which fuels the protein O-GlcNAcylation[29]. O-GlcNAcylation is critical for self-renewal and malignant transformation of T cell progenitors, thymocyte positive selection, and clonal expansion of peripheral T cells[29]. It is still unclear how TCR-activation promotes the HBP flux. In addition, Treg cells and CD4$^+$ Tconv cells program distinct cellular metabolism[55]. Tconv cells are dependent on aerobic glycolysis and glutamino-lysis to fulfill their bioenergetic and biosynthetic demand of proliferation[56]. However, Treg cells couple TCR and IL-2R signals with mTORC1-dependent lipid biosynthesis to establish their functional program[33]. It is thus unlikely that TCR-activated protein O-GlcNAcylation in Treg cells is a result of increased glucose flux. Recently, our findings in hepatocytes have indicated that Ca$^{2+}$-dependent CaMKII phosphorylates and activates OGT[57]. Ca$^{2+}$ release is induced downstream of TCR signaling and the Ca$^{2+}$-dependent serine-threonine phosphatase Calci-neurin, by dephosphorylating and mediating nuclear transloca-tion of NFAT, is crucial for Treg cell development[58,59]. It is warranted to test whether Ca$^{2+}$ signaling mediates TCR-activated O-GlcNAcylation via modulating OGT activity.

Treg cells restrain autoimmune disease, suppress inflammation, prevent organ transplant rejection, and contribute to immune

dysfunctions in some infections and cancers[60,61]. However, challenges for Treg cell immunotherapy exist as Treg cells display functional instability and heterogeneity. Our study has identified OGT and protein O-GlcNAcylation as a regulatory hub connecting TCR, FOXP3, and IL-2R to support Treg cell lineage stability and suppressive function. More importantly, pharmacological elevation of O-GlcNAcylation by inhibiting OGA augments the suppressive activity of human Treg cells, thus targeting O-GlcNAcylation can facilitate the translation of Treg cell therapy into the clinic to treat diseases such as autoimmune disorders, inflammation, and GVHD.

## Methods

**Mice**. $Foxp3^{YFP-cre}$ mice (stock number 016959) and $Foxp3^{eGFP-Cre-ERT2}$ mice (stock number 016961) were purchased from the Jackson Laboratory. $Rosa26^{Stat5b-CA}$ mice were provided by Dr. Alexander Rudensky at Memorial Sloan Kettering Cancer Center[18]. $Ogt^{fl/fl}$ Mice (Jackson Laboratory stock number 004860) were kindly provided by Dr. Xiaoyong Yang at Yale University. $Rosa26^{tdTomato}$ reporter mice (Jackson Laboratory stock number 007909) were kindly provided by Dr. Jop van Berlo at University of Minnesota. $UbcCreER+$ mice (Jackson Laboratory stock number 007001) were kindly provided by Dr. Doug Mashek at University of Minnesota. All mice were on C57BL/6 background. $Foxp3^{YFP-Cre/Y} Ogt^{fl/Y}$ male mice were used at 2-week old unless specific noted, with the same age $Foxp3^{YFP-Cre/Y} Ogt^{wt/Y}$ as control. Other mice were used at 6–12 weeks old unless specific noted. Mice were free to access water and fed on a regular chow or Tamoxifen food (Teklad, TD.130860) as indicated. All procedures were approved by the Institutional Animal Care and Use Committee at the University of Minnesota. All relevant ethical regulations for animal testing and research were complied with.

For IL-2-immune complex administration, IL-2 (eBioscience, catalog no. 14–8021–64) and JES6-1A12 (eBioscience, catalog no. 16–7022–81) were incubated at a molar ratio of 2 to 1 in PBS at room temperature for 30 min followed by 3 consecutive days i.p. injections. For female mice, each mouse received 1 μg of IL-2 and 5 μg of JES6–1A12 in 200 ul PBS at 6-week old. For male mice, each mouse received 0.5 μg of IL-2 and 2.5 μg JES6–1A12 in 100ul PBS at 2-week old.

For histology, mouse tissues were dissected and fixed in 10% buffered formalin. Sectioning and haematoxylin & eosin (H&E) staining were performed by the Comparative Pathology Shared Resource of UMN.

**Human participants**. Nonmobilized PB leukapheresis products were collected from normal adult volunteers with Food and Drug Administration (FDA)–approved/cleared apheresis instruments at Memorial Blood Center (St. Paul, MN)[45]. Written informed consent was obtained from all participants with approval from the University of Minnesota Institutional Review Board.

**Naïve T cell isolation and in vitro Treg cell induction**. Naïve $CD4^+$ T cells were purified from lymph nodes and spleen according to the protocol provided by the manufacturer (Invitrogen, catalog no. 11461D). Plates were coated with 10 μg/ml of anti-mouse CD3ε antibody (BioLegend catalog no. 100314) for 4 h at 37 °C. Before plating the cells, plates were washed once with PBS. Naïve $CD4^+$ T cells were suspended at the concentration of $2 \times 10^6$/ml in X-Vivo15 serum-free medium (Fisher Scientific, catalog no. 04–744Q) with 2 μg/ml anti-mouse CD28 antibody (BioLegend catalog no. 102112) in the presence or absence of mouse TGF-β (BioLegend catalog no. 763102). Cells were cultured for 2 days followed with retroviral transduction or for 5 days before the collection for RNA and immunoblot analysis[62].

**Treg cell purification and expansion**. $CD4^+CD25^+$ Treg cells were purified from mouse lymph nodes and spleen according to the protocol provided by the manufacturer (Invitrogen, catalog no. 11463D). Treg cells were expanded according to the Treg Expansion Kit protocol (Miltenyi Biotec, catalog no. 130–095–925). Treg cells were treated with 0.5 μM 4-Hydroxytamoxifen (4-OHT) (Millipore Sigma, catalog no. H7904) for 3 days, 10 μM TMG (CarboSynth), or 100 μg/ml cycloheximide (CHX) (Millipore Sigma, catalog no. C7698) as indicated. To determine the purity, Treg cells were assessed by flow cytometry and immunoblot analysis.

**Flow cytometry**. For surface markers, cells were stained in PBS containing 0.5% (wt/vol) BSA with relevant antibodies once at 4 °C for 30 min. For analysis of intracellular markers, cells were first fixed with Fixation/Permeabilization buffer (ThermoFisher, catalog no. 00–5123) at 4 °C for 30 min and then stained in Permeabilization Buffer (ThermoFisher, catalog no. 00–8333) with relevant antibodies at 4 °C for 30 min. For detection of Annexin V population, cells were suspended in 100 μl 1× Annexin V binding buffer after surface staining, added 5ul APC-Annexin V at room temperature for 15 min, and then added 400 μl 1× Annexin V binding buffer for flow cytometry measurement. For pY-STAT5 signaling detection, lymphocytes were stimulated with IL-2 (0.5 ug/ml in PBS) (BioLegend, catalog no. 575402) at 37 °C for 20 min, fixed with 3% Paraformaldehyde (formaldehyde)

aqueous solution (Electron Microscopy Sciences, catalog no. 15710) at 1:1 ratio at room temperature for 10 min, washed with PBS, resuspended in 100% cold methanol at 4 °C for 10 min and finally stained with antibodies for FACS. For intracellular cytokine staining, cells were stimulated with PMA (50 ng/ml, Sigma, catalog no. P1585) and Ionomycin (500 ng/ml, Sigma, catalog no. I3909), together with GolgiPlug (BD Biosciences, catalog no. 555029) for INFy, IL-5, IL-13, and IL-17A or GolgiStop (BD Biosciences, catalog no. 554724) for IL-4 at 37 °C for 5 h before being stained. Antibody information were shown in Supplementary Table 2. Flow cytometry data were acquired on BD Fortessa X-20 and analyzed with FlowJo (v10.5.3). Gating strategies were shown in Supplementary Fig. 11.

**Suppression assay**. Naïve human PB tTreg ($CD4^+CD25^+CD127^-CD45RA^+$) were sort-purified from PB mononuclear cells (PBMNCs) (Ficoll-Hypaque, Amersham Biosciences) in a two-step procedure in which $CD25^+$ cells were initially enriched from PBMNCs by AutoMACS (PosselD2) with GMP grade anti-CD25 microbeads (Miltenyi Biotec). $CD25^{high}$ cells were stained with CD4, CD8, CD25, CD127 and CD45RA and sorted via FACSAria as $CD4^+$, $CD8^-$, $CD25^{high}$, and $CD127^-CD45RA^+$. Note that the bead-bound and fluorochrome-conjugated anti-CD25 antibodies recognize different epitopes (Miltenyi Biotec), followed by sorting on a FACSAria (BD Biosciences).

Purified naïve tTreg were stimulated with a K562 cell line engineered to express CD86 and the high-affinity Fc receptor (CD64) (37) (2:1 tTreg/KT), which had been irradiated with 10,000 cGy and incubated with anti-CD3 mAb (Miltenyi Biotec). In some experiments, tTregs were stimulated with KT64/86 cells that were preloaded, irradiated, and frozen (1:1 tTreg/KT). Naive tTreg were cultured in X-Vivo-15 (BioWhittaker, Walkersville, MD) media supplemented with 10% human AB serum (Valley Biomedical), Pen/Strep (Invitrogen), and n-acetyl cysteine (USP). Recombinant IL-2 (300 IU/mL; Chiron, Emeryville, CA) was added on day 2 and maintained for culture duration. Cultures were maintained at $0.25 \times 10^6$ to $0.5 \times 10^6$ viable nucleated cells/ml every 2 to 3 days. On day 14, tTreg and $CD4^+$ Teff were aliquoted and frozen. When needed, frozen tTreg were thawed and re-stimulated with anti-CD3/CD28 mAb-coated Dynabeads (Life Technologies, Carlsbad, CA) at 1:3 (cell to bead) ratios in the presence of recombinant IL-2 (300U/ml). OGA inhibitor (10 μM) was added to re-stimulated tTreg cultures on day 0 and day 4. tTreg was cultured a total of 7 days following re-stimulation and were then harvested for flow phenotyping and suppressive function.

The in vitro suppressive capacity of expanded tTregs was assessed with a CFSE (Carboxyfluorescein succinimidyl ester) inhibition assay[63,64]. Briefly, PBMNCs were purified, labeled with CFSE (Invitrogen), and stimulated with anti-CD3 mAb-coated beads (Dynal) ± cultured tTreg (1:2 to 1:8 tTregs/PBMC). On day 4, cells were stained with antibodies to CD4 and CD8 and suppression was determined from the Division Index (FlowJo, TreeStar). tTregs suppressed CD4 and CD8 T cell responses equivalently and only CD8 data were presented.

**Cell culture and transfection**. HEK 293 T cells (ATCC) were cultured in DMEM with 10% fetal bovine serum (FBS). TMG (CarboSynth, 10 μM), ST045849 (Tim-Tec, 100 μM), MG132 (Cayman, 20 μM), and cycloheximide (Sigma, 100 μg/ml) were treated as indicated. The mouse FOXP3-Myc/DDK plasmid was purchased from OriGene (MR227205). Point mutants of FOXP3 were generated with the QuikChange XL II Site-Directed Mutagenesis Kit (Agilent). pCMV-Myc-human OGT and 3xFlag/2xMyc-OGA were kindly provided by Dr. Xiaochun Yu at University of Michigan and Dr. Xiaoyong Yang at Yale University, respectively. FOXP3-Myc/DDK was subcloned into the 3xFlag/2xMyc empty vector to construct the 3xFlag/2xMyc-FOXP3-Myc/DDK plasmid. IL-2 promoter luciferase (#12194) and pIS1 (#12179) were from Addgene. Cells were transfected with FuGENE HD (Promega) according to manufacturer's instructions.

**Mass spectrometry**. 3xFlag/2xMyc-FOXP3-Myc/DDK and Myc-OGT were co-transfected into $5 \times 15$cm-dishes of 293 T cells and purified by immunoprecipitation with M2 Flag beads (Sigma) followed by 3xFlag peptide (Sigma) elution according to established procedures[25]. IP eluates were denatured in 0.2% Rapigest SF (Waters), reduced with 5 mM DTT, alkylated with 10 mM Iodoacetamide, and finally digested overnight at 37 °C with 5% (w:w) sequencing grade Trypsin (Promega). Digests were acidified with formic acid for 30 mins to degrade the Rapigest and peptides were then recovered and desalted with $C_{18}$ OMIX tips (Agilent).

Tryptic peptides were analyzed by on-line LC-MS/MS using an Orbitrap Fusion Lumos (Thermo) coupled with a NanoAcquity UPLC system (Waters). Peptides were separated over a 15 cm × 75 μm ID 3 μm C18 EASY-Spray column (Thermo). Precursor ions were measured from 350 to 1800 m/z in the Orbitrap analyzer (resolution: 120,000; AGC: 4.0e5). Ions charged 2 + to 8 + were isolated in the quadrupole (selection window: 1.6 m/z units; dynamic exclusion window: 30 s; MIPS Peptide filter enabled), fragmented by EThcD (Maximum Injection Time: 250 ms, Normalized Collision Energy: 25%) and measured in the Orbitrap (resolution: 30,000; AGC: 5.0e4). The cycle time was 3 s.

Peaklists were generated using PAVA (UCSF) and searched using Protein Prospector 5.23.0 against the human SwissProt database (downloaded 9/6/2016) and a randomized concatenated database with the addition of the recombinant FoxP3 recombinant sequence. Cleavage specificity was set as Trypisn allowing 2

miscleavages. Carbamidomethylation of Cys was set as a constant modification and two of the following variable modifications were allowed per peptide: acetylation of protein N-termini, oxidation of Met, oxidation and acetylation of protein N-terminal Met, cyclization of N-terminal Gln, protein N-terminal Met loss, protein N-terminal Met loss and acetylation, acetylation of Lys, phosphorylation of Ser, Thr, Tyr, HexNAc on Asn within the N glycan motif (NXST), HexNAc on Ser, Thr, Tyr, and PhosphoHexNAc on Ser or Thr. Precursor mass tolerance was 20ppm and fragment mass tolerance was 30 ppm. Phosphorylated and HexNAcylated peptides were manually verified.

**RNA and real-time PCR**. Total RNA was extracted using RNeasy Plus Universal Kits (QIAGEN, catalog no. 73404). cDNA was reverse transcribed (BIO-RAD, catalog no. 170–8891) and amplified with SYBR Green Supermix (BIO-RAD, catalog no. 172–5124). All data were normalized to the expression of *18 S*. Primer sequences were shown in Supplementary Table 1.

**Immunoprecipitation and Western blotting**. Anti-O-GlcNAc (Abcam, catalog no. ab2739, 1:200), anti-FOXP3 (eBioscience, catalog no. 56–7773–82, 1:100), anti-c-Myc (SANTA CRUZ, catalog no. sc-40, 1:500), anti-Flag (Millipore Sigma, catalog no. F3165, 1:1000), anti-β-Actin (Millipore Sigma, catalog no. A5441, 1:1000) antibodies were used. For Immunoprecipitation, whole cell lysates were precipitated by anti-FLAG M2 Affinity Gel (Millipore Sigma, catalog no. A2220). Cells were lysed in buffer containing 1% Nonidet P-40, 50 mM Tris 3 HCl, 0.1 mM EDTA, 150 mM NaCl, proteinase inhibitors and TMG. Equal amounts of protein lysates or immunoprecipitation samples were electrophoresed on 8% SDS-PAGE gels and transferred to PVDF or Nitrocellulose membranes. Primary antibodies were incubated at 4 °C overnight. Western blotting was visualized by peroxidase conjugated secondary anti-bodies and ECL chemiluminescent substrate or by 800CW infrared fluorescent IgG secondary antibodies imaged on a Bio-Rad ChemiDoc Imaging System or a LI-COR Odyssey 9120 infrared imager, respectively. Uncropped Western blot images can be found in Supplemental Fig. 12.

**Retroviral transduction**. STAT5A, constitutively active (CA)- STAT5A, and O-GlcNAc-deficient CA-STAT5A-T92A retroviral expression plasmids were kindly provided by Dr. Richard Moriggl at University of Veterinary Medicine, Vienna. Wildtype and mutant FOXP3 were constructed into the pMSCV-IRES-eGFP retroviral vector. Retrovirus were packaged in 293FT cells (ThermoFisher, R70007) and viral supernatants were harvested for use. Treg and Naïve T cells were cultured 48 h, then spin infected with viral supernatants (2500 rpm, room temperature for 2 h) in the presence of 8 μg/ml polybrene[65]. Cells were cultured for another 3 days.

**RNA-seq**. Treg cells were isolated from female *Foxp3^{YFP-Cre/wt}Ogt^{fl/fl}* and *Foxp3^{YFP-Cre/wt}Ogt^{wt/fl}* mice using CD4 selection beads (ThermoFisher, 11461D) followed by sorting on YFP expression. RNA was extracted using the RNeasy Plus Micro Kit (QIAGEN, catalog no. 74034) and subjected to library construction and 2x50 paired-end sequencing on a HiSeq 2500 (Illumina) at the University of Minnesota Genomics Center. Sequencing reads were trimmed (Trimmomatic v0.33), quality control checked (FastQC), mapped (bowtie2 v2.2.4.0), and quantified for gene expression (Cuffquant). Differentially expressed genes were determined by EdgeR. Pathway analyses and upstream regulator analyses were performed using Ingenuity Pathway Analysis (Qiagen).

**Statistical analyses**. Results are shown as mean ± SEM. The comparisons were carried out using two-tailed Student's *t* test, one-way ANOVA followed by multiple comparisons with the Tukey adjustment, two-way ANOVA followed by multiple comparisons with the Tukey adjustment, or Kaplan-Meier Analysis as indicated using GraphPad Prism 7.

## Data availability
RNA Sequence data that support the findings of this study have been deposited in Gene Expression Omnibus with the accession number GSE116758. Mass spectra data can be accessed on MS-Viewer[66] with the key: artnlq2gpq.

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

## Acknowledgements

We thank Dr. Hu Zeng for discussions and suggestions, Dr. Xiaoyong Yang for providing the *Ogt^{fl/fl}* mice, Dr. Richard Moriggl for providing STAT5 vectors, Dr. Alexander Rudensky for providing the *Rosa26^{Stat5b-CA}* mice, Dr. Jop van Berlo for providing the *Rosa26^{tdTomato}* Cre-reporter mice, and Dr. Doug Mashek for providing the *Ubc-Cre/ERT2^+* mice. We thank the UMN Flow Cytometry Resource for cell sorting, Juan E. Abrahante Lloréns from University of Minnesota Informatics Institute for bioinformatic analysis. This work was supported by NIH grant R01 AI139420, R21 AI140109, and American Diabetes Association grant 1–18-IBS-167 to H.-B.R., R37 AI39560 to K.A.H., R01 AI124512 to M.A.F., R01 HL11879 and P01 CA065493 to B.R.B., S10 OD010731 to L.E.B., and a grant from the Adelson Medical Research Foundation to A.L.B. The Thermo Scientific Fusion Lumos was funded by the UCSF Program for Breakthrough Biomedical Research (PBBR).

## Author contributions

B.L. designed and performed experiments, analyzed and interpreted data, and wrote the manuscript; O.C.S. designed and performed flow cytometry and analyzed and interpreted data; S.S. and K.L.H. performed flow cytometry and suppression assay on human tTreg cells. J.C.M., A.L.B., and L.E.B. performed O-GlcNAc mass spectrometry. B.R.B. contributed to data interpretation and manuscript editing. M.A.F. and K.A.H contributed to experimental design and edited the manuscript; H.-B.R. conceived and supervised the project and wrote the manuscript.

## Additional information

**Competing interests:** The authors declare no competing interests.

