## [Peer Review File · Nature Communications]

Reviewers' comments:

Reviewer #1 (Treg, TCR signaling)(Remarks to the Author):

In this study, the authors have examined the effects of O-GlcNAcylation on Treg development and function in vivo, by deleting the gene OGT with a FoxP3-driven Cre. They show that these mice (at least male mice, which are fl/Y) develop an autoimmune phenotype reminiscent of the Scurfy phenotype seen in certain other Treg-deficient models. They then do some flow cytometry characterization of the phenotype of Treg from these mice. In addition, they perform RNAseq to probe the underlying transcriptional differences between WT and OGT-deficient Treg. While there are some interesting data presented here, the manuscript is not coherent or thorough, with somewhat superficial analyses of several different aspects of Treg phenotype and function. Specific critiques are below.

1. Looking at Fig. 1 A and B, and comparing them to supplementary Fig. 1A and B, it looks like thymic Treg and peripherally induced Treg display different levels of OGT and O-GlcNAcylation, but this possibility is not addressed directly by the authors. Do the authors believe that peripherally induced Treg are more stable based just on O-GlcNAcylation (this seems to be contrary to what is currently thought, based on the literature)?
2. In Fig. 1 J-M, the authors show in multiple ways that loss of O-GlcNAcylation leads to destabilization of FoxP3 protein, however they do not make any comment about the actual cell numbers attained in the culture system.
3. In Fig. 2, the authors show severe autoimmunity followed by characterization of two-week-old mice, but they do not show or discuss any other lymphocyte compartment, even though they see severe autoimmunity. It would be interesting to see how other lymphocyte compartments are affected (in terms of frequency and absolute cell counts). In addition, are serum Ig levels affected in the mice with autoimmunity?
4. Importantly, FoxP3-YFP-Cre mice are known to be leaky, at least with some floxed alleles (e.g. CD28; Franckaert et al. (2015) Immunol. Cell Biol.). Since the authors seem to have flow cytometry-compatible OGT antibody, it would be critical for the authors to show OGT levels on other lymphocyte cell types like CD4, CD8, B cells etc. Authors need to use an inducible mouse model like tamoxifen-inducible model (which is not detectably leaky) to conclusively prove the existence of ex-Treg.
5. In Fig. 4 and supplementary Fig. 4, the authors claim that there is a difference in certain functional markers between WT and OGT KO Treg cells, but there is not conclusive evidence that the markers looked at are directly related to suppressive function in vivo (or even in vitro). The authors also need to look at this phenotype in more detail. For example, in OGT het female mice, the authors could easily gate on OGT-sufficient and OGT-deficient Treg and compare expression of the functional markers.
6. In Fig. 5, based on their gene expression profiling, the authors suggest a role for Blimp-1 in regulation of Treg by O-GlcNAcylation, but the authors do not follow-up this lead in the manuscript. The authors also do not comment specifically on whether FoxP3-regulated genes are affected. For example, GlcNAcylation may affect FoxP3 binding to DNA.
7. Based on Fig. 5 and Supplementary Fig. 5, the authors try to conclude that OGT leads to effects on Stat5 target genes independent of phosphorylation, but they do not explain how this might work. One possibility that authors need to test is that localization of Stat5 in the nucleus is affected.

8. In Fig. 7, the authors randomly move to a human system without testing the same hypothesis on Treg from the mouse model (see above).

9. Although the authors have focused on the IL-2 receptor signaling pathway, based on their gene expression analyses, they need to comment on TCR signaling in these mutant Treg. Some basic *in vitro* characterization to look at pS6 and pAkt would definitely add to a better understanding of how O-GlcNAcylation in Treg can affect function. In the final discussion, the authors claim that (line 354) that they are connecting TCR, FoxP3 and IL-2R, although TCR signaling has not really been examined.

Reviewer #2 (Treg, epigenetic/DNA methylation)(Remarks to the Author):

The study by Dr. Liu et al demonstrates that FOXP3 protein can be modified and stabilized by O-GlcNAcylation in HEK293T cells and Treg cells. Using genetically modified mice with Treg cell-specific deletion of OGT (Foxp3YFP-Cre/YOgtfl/Y mice), the authors evaluated whether OGT regulates mature Treg cell function *in vivo* and their results suggest that O-GlcNAc-deficient Treg cells develop normally but display modestly reduced FOXP3 expression. Using another genetically modified mouse model, the Foxp3YFP-Cre/YOgtfl/YRosa26Stat5b-CA/wt mice with STAT5B-CA overexpression specifically in OGT-deficient Treg cells, they further evaluated whether STAT5 activation could restore Treg cell function in Foxp3YFP-Cre/Y Ogtfl/Y mice and their results indicate that overexpression of a constitutively active form of STAT5 partially rescues Treg-cell dysfunction and systemic inflammation. The authors conclude that protein O-GlcNAcylation is essential for lineage stability and effector function in Treg cells.

The strengths of this study are the generation of adequate genetic tools-mice with conditional OGT ablation in specific foxp3 expression Treg cells, and mice with STAT5B-CA overexpression in OGT-deficient Treg cells, which can demonstrate unequivocally that the role of OGT and STAT5 in regulation of Treg *in vivo*. However, there are several issues worth addressing:

Major remarks:

1. Figure 1A and 1B show the expression level of O-GlcNAc or OGT in CD4+CD25+ Treg cells. The Treg markers (such as CD25, CTLA-4, GITR and FOXP3) used represent general T-cell activation markers, and not true Treg specific markers. Foxp3 is considered as the most reliable marker for Treg cells [Immunity. 2005; 22: 329–41]. Evaluating these two proteins in Treg cells identified with CD4+Foxp3+ or CD4+CD25+FOXP3+ cells would strengthen the results.

2. A previous study suggested that O-GlcNAc is important for T cell activation and that expression of O-GlcNAc is increased in human T cells activated with anti-CD3/CD28 beads [J Immunol. 2016 Oct 15; 197(8): 3086–3098]. The authors describe "TCR activation further promoted protein O-GlcNAcylation in Treg cells *ex vivo* (Figure 1C)." Were the cell lysates used for analysis of O-GlcNAc level by immunoblot in Figure 1C from sorted Treg cells or whole cell stimulated with anti-CD3/ CD28 beads? How many samples and how many independent experiments have been done for immunoblot experiments? The protein markers (ladder) are missing in immunoblot results shown in Fig 1C and Supplement 1B.

3. Figures 1J-1M show reduced MFI of foxp3 in CD4+CD25+ and CD4+CD25+ Treg frequency in Ubc-Cre/ERT2+Ogtfl/Y cells treated with 4-OHT. What is the frequency of Treg cells identified with CD4+Foxp3+ or CD4+CD25+FOXP3+ cells?

4. IL4, IL13, and IL5 are produced by Th2 effector T cells. Differential expression levels of Th2 cytokines are seen in Th2 cells in vivo. A previous study suggested that increased GATA-3 expression is essential for IL-5 and IL-13 production by Th2 cells, but not IL-4 [Nat Immunol. 2004 Nov; 5(11): 1157-65]. The results in Figure 2J show increased GATA3 expression in Foxp3YFP-Cre/Y Ogtfl/Y mice. What would the IL-13 and IL-5 expression levels be in effector CD4+ T cells in Foxp3YFP-Cre/Y Ogtfl/Y mice, although slightly elevated IL4 expression has been shown in Figure 2L?
5. Figure 3I and 3J show the expression of Th1 T-bet in OGT-deficient ex-Treg cells. What would the expression level be in other Th cells such as Th17 TF ROR γ and Th2 TF GATA3?
6. A previous study demonstrated the presence of many Treg IL-2 dependent genes, such as Klrp1, IL10, Blimp-1 et al [Immunity. 2009 Feb 20; 30(2):204-17]. The results in Figure 5E-G show KLRG1 expression for evaluation the responsiveness of Treg cells to IL-2. It would be interesting to examine the other IL-2 dependent targets in Treg cells.
7. Comparisons of MFI of Foxp3 expression and CD25+FOXP3+ Treg frequency from Foxp3YFP-Cre/YOgtfl/YRosa26Stat5b-CA/wt mice (STAT5B-CA overexpression OGT-deficient Treg cells) and Foxp3YFP-Cre/YOgtfl/Y mice (OGT-deficient Treg cells) are shown in Figures 6D and 6E. To demonstrate the rescue dysfunction of OGT deficient Treg cells, it would be more explicit to include the Treg analysis from Foxp3YFP-Cre/YOgtwt/Y mice.
8. For the suppression function of human Treg cells, in Figure 7E, please add the representative flow plots showing the cell proliferation of responder cells based on the dilution of fluorescence intensity of CFSE of gated cells.

Reviewer #3 (Post-translational modification, glycosylation)(Remarks to the Author):

Ruan, Hogquist and co-workers report on the function of O-GlcNAc in maintenance of lineage stability and effector function in Treg cells. The major finding is that without O-GlcNAc through knockdown or inhibition of the enzyme that installs it, OGT, Treg cells are dysregulated and promote systemic inflammation as observed through a scurfy phenotype in mice. This phenotype can be partially rescued by the over expression of STAT5, which is downstream of IL-2. This examination of the regulatory importance of the O-GlcNAc modification in a previously unexamined T cell subtype will be of significant interest to glycoimmunologists and immunologists alike. The results described here underlie a need for better understanding of O-GlcNAc individual cell types within the immune system to better regulate the immune system.

While the authors report an impressive functional outcome from inhibition of OGT in Treg cells, the authors describe an incomplete molecular mechanism through FOXP3 and STAT5 and linkage of the TCR and IL-2 pathways. The connection of O-GlcNAc on FOXP3 to the phenotype observed in OGT-deficiency is tenuous at best. The authors find that FOXP3 is moderately stabilized by O-GlcNAc, but very limited connection of the glycosylation on FOXP3 is related back to the broader mechanistic outcome. Most of these studies rely on global manipulation of O-GlcNAc levels through genetic or chemical means. Knockdown of OGT will affect a number of functions including destabilization of FOXP3 as the authors show. The best approach to relate function back to the specific glycosites is to trace down the glycosites for mutation and correlate the mutants with reduction of the phenotype in line with global methods to regulate O-GlcNAc that the authors used. Otherwise, the molecular outcomes from OGT knockdown in Treg cells may truly derive from mechanisms through other

proteins. Further examination of O-GlcNAc sites on FOXP3 would add to whether or not it plays a direct role in mediating auto-immune regulation in Treg cells. The authors do examine constitutively active STAT5 via a mutagenesis approach; however, these studies are uniformly performed in OGT-deficient mice, leading to a lack of clarity about STAT5 rescue being related to O-GlcNAc (what is the rationale for knocking out an O-GlcNAc site on STAT5 in O-GlcNAc deficient mice?). Nonetheless, the manuscript describes a surprising and intriguing dysregulation of Treg cells without O-GlcNAc that adds to the growing importance of O-GlcNAc in the immune system.

Minor points

- Rescue of mice at week 2 (Figure 6), how does it longitudinally correlate to these mice at week 4 and their longevity?
- Is O-GlcNAcylation stabilizing FOXP3 after TCR activation or simply stabilizing it as it is being expressed more?

Responses to Reviewers' Comments:

We thank the reviewers very much for their constructive critiques and comments. We have specifically addressed each of the points raised by the reviewers with additional experiments and text changes. Specifically, we have provided more data on functional characterization of FOXP3 O-GlcNAcylation, on immune cell subsets in *Foxp3-Ogt* KO mice, and on Treg cell dysfunction caused by OGT deficiency. These new additions that have been incorporated into the revised manuscript further strengthen and support the results and conclusions.

Reviewer #1:

1. Looking at Fig.1 A and B, and comparing them to supplementary Fig. 1A and B, it looks like thymic Treg and peripherally induced Treg display different levels of OGT and O-GlcNAcylation, but this possibility is not addressed directly by the authors. Do the authors believe that peripherally induced Treg are more stable based just on O-GlcNAcylation (this seems to be contrary to what is currently thought, based on the literature)?

Response: We apologize for any confusion, but we did not intend to suggest that peripherally induced Treg (iTreg) cells have higher O-GlcNAcylation levels or are more stable than thymic Treg cells. Fig.1 A and B showed that thymic Treg cells displayed similar levels of O-GlcNAcylation and OGT compared to Naïve T cells in mice. Data in Supplementary Fig.1A and B, however, were obtained from cells cultured and treated for 5 days ex vivo. We think it is not appropriate to compare the O-GlcNAcylation level of anti-CD3/CD28 beads-treated CD4⁺CD25⁻ cells with that of naïve T cells in vivo and then use it to make conclusions about central and peripheral Treg cells. Future experiments are required to determine if protein O-GlcNAcylation is a mechanism for differential stability between thymic and induced Treg cells.

2. In Fig. 1 J-M, the authors show in multiple ways show that loss of O-GlcNAcylation leads to destabilization of FoxP3 protein, however they do not make any comment about the actual cell numbers attained in the culture system.

Response: Similar number of CD4⁺CD25⁺ cells isolated from *Ubc-CreER⁺Ogt^{fl/Y}* mice were used for the treatment of vehicle or 4-OHT. For flow cytometry, around 250,000 total cells per sample were analyzed and around 3,000 CD4⁺FOXP3⁺ Treg cells were obtained. These details have now been added in the revised figure legend. Data shown in the new Fig. 2A-D are all shown on a per cell basis or as percentage. As shown in Fig. 2C, D, the percentage of CD4⁺FOXP3⁺ Treg cells was significantly downregulated when OGT was deleted upon 4-OHT treatment.

3. In Fig. 2, the authors show severe autoimmunity followed by characterization of two-week-old mice, but they do not show or discuss any other lymphocyte compartment, even though they see severe autoimmunity. It would be interesting to see how other lymphocyte compartments are affected (in terms of frequency and absolute cell counts). In addition, are serum Ig levels affected in the mice with autoimmunity?

Response: We thank the reviewer for this helpful suggestion. In addition to the number and activity of CD4⁺ and CD8⁺ T cells (shown in the new Fig. 3), we have now also included the characterization of B cells of two-week-old mice. As shown in Supplementary Fig.3C, there was a significant increase of the B cell population in *Foxp3^{YFP-Cre/Y}Ogt^{fl/Y}* KO mice, when compared to *Foxp3^{YFP-Cre/Y}Ogt^{wt/Y}* mice. Furthermore, levels of serum IgG, IgM, and free *Kappa* and *Lambda* chains were also dramatically upregulated in *Foxp3^{YFP-Cre/Y}Ogt^{fl/Y}* KO mice (Supplementary Fig.3D).

4. Importantly, FoxP3-YFP-Cre mice are known to be leaky, at least with some floxed alleles (e.g. CD28; Franckaert et al. (2015) Immunol. Cell Biol.). Since the authors seem to have flow cytometry-compatible OGT antibody, it would be critical for the authors to show OGT levels on other lymphocyte cell types like CD4, CD8, B cells etc. Authors need to use Inducible mouse model like tamoxifen-inducible model (which is not detectably leaky) to conclusively prove the existence of ex-Treg.

Response: We agree with the reviewer that limitations exist when using the *Foxp3*^{YFP-Cre} line. We performed flow cytometry using a pan-O-GlcNAcylation-specific antibody and found that global protein O-GlcNAcylation was downregulated only in CD4⁺FOXP3⁺ Treg cells, but not CD4⁺FOXP3⁻ T cells, CD8⁺ T cells, B cells, or NK cells (Supplementary Fig. 3A), suggesting that the potential leak of *Foxp3*^{YFP-Cre} was transient or on a small scale.

As suggested by the reviewer, to conclusively prove the existence of ex-Treg cells, we used a tamoxifen-inducible model by generating the *Foxp3*^{eGFP-Cre-ERT2/Y} *Ogt*^{fl/Y} *Rosa26*^{tdTomato/wt} mice. Tamoxifen-containing diet induced Treg cell-specific ablation of protein O-GlcNAcylation (Supplementary Fig. 5A) and progressive systemic inflammation in *Foxp3*^{eGFP-Cre-ERT2/Y} *Ogt*^{fl/Y} *Rosa26*^{tdTomato/wt} mice (Supplementary Fig. 5B-E). Gating on the CD4⁺GITR⁺tdTomato⁺ population, we found that more FOXP3⁺tdTomato⁺ ex-Treg cells emerged in *Foxp3*^{eGFP-Cre-ERT2/Y} *Ogt*^{fl/Y} *Rosa26*^{tdTomato/wt} KO mice than in *Foxp3*^{eGFP-Cre-ERT2/Y} *Ogt*^{wt/Y} *Rosa26*^{tdTomato/wt} controls (Fig. 4G). These ex-Treg cells were prone to express the Th transcription factors including T-BET and GATA3 (Fig. 4H, I). These results further support the conclusion that OGT-mediated protein O-GlcNAcylation contributes to the maintenance of Treg lineage stability.

5. In Fig. 4 and supplementary Fig. 4, the authors claim that there is a difference in certain functional markers between WT and OGT KO Treg cells, but there is not conclusive evidence that the markers looked at are directly related to suppressive function in vivo (or even in vitro). The authors also need to look at this phenotype in more detail. For example, in OGT het female mice, the authors could easily gate on OGT-sufficient and OGT-deficient Treg and compare expression of the functional markers.

Response: Using the constitutive *Foxp3*^{YFP-Cre/Y} *Ogt*^{fl/Y} and inducible *Foxp3*^{eGFP-Cre-ERT2/Y} *Ogt*^{fl/Y} mice, we specifically deleted OGT in Treg cells and these mice progressively developed scurfy phenotypes (Fig. 3 and Supplementary Fig. 5), strongly indicating a loss of Treg suppressive function in vivo. However, an in vitro assay of Treg's ability to abrogate lymphocyte proliferation did not reveal any difference between OGT-sufficient and OGT-deficient cells (data not shown), indicating that OGT is only indispensable for part of the immune functions regulated by Treg cells and/or compensatory mechanisms may be evolved (Fig. 6A). Similar observations have been reported in many other loss-of-function models, where Treg cell-dysfunction in vivo was accompanied by intact suppressive function in vitro^{2, 3, 4, 5}. We thus examined the expression of multiple effector molecules including KLRG1, PD-1, CD73, and CD103 in Treg cells. KLRG1-expressing cells are a subset of short-lived terminally differentiated Treg cells with an effector signature⁶. PD-1 expression in Treg cells prompts FOXP3 expression and enhances suppressive activity⁷. CD73 on Treg cells converts AMP to adenosine to mediate the immune suppression⁸. CD103, which is necessary for the retention of Treg cells at the site of inflammation, identifies the most potent subset of Treg cells^{9, 10}. The loss of these effector molecule-expressing populations in *Foxp3*^{YFP-Cre/Y} *Ogt*^{fl/Y} mice (Fig. 5A-D) demonstrates that OGT is required for the development of effector Treg cells. In the revised manuscript, we also have removed sentences indicating OGT knockout is directly related to defective suppressive function in vivo.

As the reviewer suggested, we also looked at these phenotypes in females. As shown in Fig. 5E-I (originally Fig 4E-I), we compared YFP⁺ OGT-deficient and YFP⁻ OGT-sufficient Treg cells in heterozygous female *Foxp3*^{YFP-Cre/wt} *Ogt*^{fl/fl} mice and observed a significant downregulation of effector Treg populations when OGT was deficient. In addition, we also compared YFP⁺ Treg cells between *Foxp3*^{YFP-Cre/wt} controls and *Foxp3*^{YFP-Cre/wt} *Ogt*^{fl/fl} KO females, and similar loss of effector Treg cells was observed in KO mice (Supplementary Fig. 6D-H).

6. In Fig. 5, based on their gene expression profiling, the authors suggest a role for Blimp-1 in regulation of Treg by O-GlcNAcylation, but the authors do not follow-up this lead in the manuscript. The authors also do not comment specifically on whether FoxP3-regulated genes are affected. For example, GlcNAcylation may affect FoxP3 binding to DNA.

Response: In the new Fig. 6, we first used a BLIMP-1-regulated gene profile as a maker for effector Treg cells, because BLIMP-1 is common to all eTreg cells¹¹. BLIMP-1-upregulated genes were enriched in OGT-sufficient Treg cells, while BLIMP-1-downregulated genes were enriched in OGT-deficient Treg cells (Fig. 6B, C), suggesting that OGT maintains a transcriptional program similar to that of BLIMP-1⁺ eTreg cells. We then followed up by examining the regulation of BLIMP-1 expression by OGT. Even though the expression of BLIMP-1 was comparable between OGT-sufficient and OGT-deficient Treg cells, IL-2-stimulated expansion of BLIMP-1⁺ eTreg cells was absent when OGT was deleted (Fig. 6G, J).

FOXP3-regulated genes were not enriched in the differentially expressed genes between OGT-sufficient and OGT-deficient Treg cells (data not shown). To determine the effect of O-GlcNAcylation on FOXP3 transcriptional activity, we performed a Forkhead-responsive element luciferase assay and found that OGT overexpression enhanced the suppressive activity of FOXP3 (Supplementary Fig. 2F). Interestingly, the O-GlcNAc-deficient FOXP3 was less responsive to the activation by OGT. While these data suggest that O-GlcNAcylation has an effect on FOXP3 transcriptional activity, future experiments are needed to determine if FOXP3-binding to DNA is directly controlled by O-GlcNAcylation.

7. Based on Fig. 5 and Supplementary Fig. 5, the authors try to conclude that OGT leads to effects on Stat5 target genes independent of phosphorylation, but they do not explain how this might work. One possibility that authors need to test is that localization of Stat5 in the nucleus is affected.

Response: In an elegant study by Moriggl and colleagues, O-GlcNAcylation of STAT5 has been shown to control tyrosine phosphorylation, tetramer formation, and transcriptional activity¹². However, they also showed in the supplementary data that the STAT5-T92A mutant still can be tyrosine phosphorylated and retains DNA-binding ability, in response to EPO stimulation. As the reviewer suggested, we also examined the localization of STAT5 and found that the T92A mutation did not affect subcellular localization of STAT5 (Supplementary Fig. 8C). Future studies are warranted to investigate phosphorylation-independent regulation of STAT5 by O-GlcNAcylation.

8. In Fig. 7, the authors randomly move to a human system without testing the same hypothesis on Treg from the mouse model (see above).

Response: In the new Fig. 8, we intended to test whether enhancing protein O-GlcNAcylation could augment the suppressive function of Treg cells. We first isolated CD4⁺CD25⁺ Treg cells from mice and treated them with the OGA inhibitor TMG to elevate global protein O-GlcNAcylation. Consistent with our hypothesis that O-GlcNAcylation maintains Treg lineage stability and effector function, we observed that TMG treatment upregulated FOXP3 expression

on a per-cell basis, increased Treg cell number, activated the expression of STAT5-target genes and effector molecules (Fig. 8A-F). We then moved to human cells. Even though FOXP3 expression was not affected by TMG, it increased STAT5-target gene expression and promoted the suppressive activity of human Treg cells (Fig. 8G-K). Collectively, these data in both mice and human cells reveal that O-GlcNAcylation signaling plays a fundamental role in supporting the suppressive program of Treg cells and targeting O-GlcNAcylation can facilitate the translation of Treg cell therapy into the clinic to treat diseases such as autoimmune disorders, inflammation, and GVHD.

9. Although the authors have focused on the IL-2 receptor signaling pathway, based on their gene expression analyses, they need to comment on TCR signaling in these mutant Treg. Some basic in vitro characterization to look at pS6 and pAkt would definitely add to a better understanding of how O-GlcNAcylation in Treg can affect function. In the final discussion, the authors claim that (line 354) that they are connecting TCR, FoxP3 and IL-2R, although TCR signaling has not really been examined.

Response: We examined TCR signaling as suggested by the reviewer and found that the intensity of S6 phosphorylation but not AKT S473 phosphorylation was decreased in OGT-deficient Treg cells (Supplementary Fig. 6A), suggesting a mTORC1-specific defect. Earlier studies have shown that T cells express and upregulate O-GlcNAcylation upon immune activation^{13, 14}. Key downstream mediators of TCR signaling, including ZAP70, NF- κ B and NFAT, can be modified by O-GlcNAcylation^{15, 16, 17, 18}. Consistent with findings in conventional T cells, here we showed that TCR activation promoted protein O-GlcNAcylation in Treg cells as well. Loss of protein O-GlcNAcylation in Treg cells affected FOXP3 stability, IL-2 signaling, and mTORC1 activation. Therefore, we propose that protein O-GlcNAcylation links TCR signaling to FOXP3 and IL-2R. Nonetheless, a complete understanding of TCR-regulated O-GlcNAc proteome and the feedback regulation of TCR signaling by O-GlcNAcylation requires future investigations.

Reviewer #2:

1. Figure 1A and 1B show the expression level of O-GlcNAc or OGT in CD4⁺CD25⁺ Treg cells. The Treg markers (such as CD25, CTLA-4, GITR and FOXP3) used represent general T-cell activation markers, and not true Treg specific markers. Foxp3 is considered as the most reliable marker for Treg cells [Immunity. 2005;22:329–41]. Evaluating these two proteins in Treg cells identified with CD4⁺Foxp3⁺ or CD4⁺CD25⁺FOXP3⁺ cells would strengthen the results.

Response: We agree with the reviewer that FOXP3 is the most reliable marker for Treg cells. In Fig 1A and B, the CD4⁺CD25⁺FOXP3⁺ population was actually gated for the analysis of OGT and O-GlcNAcylation expression. We are sorry for the typo and have corrected this in the revised figure legend. In addition, we used FOXP3, *Foxp3*^{YFP-Cre}, or *Foxp3*^{eGFP-Cre-ERT2} to identify Treg cells throughout the paper, including but not limited to Fig. 1A-C, Fig. 2A-D, G, H, Fig. 4, Fig. 5, Fig. 6E-J, Fig. 7D-H, and Fig. 8A-C, G, H.

2. A previous study suggested that O-GlcNAc is important for T cell activation and that expression of O-GlcNAc is increased in human T cells activated with anti-CD3/CD28 beads [J Immunol. 2016 Oct 15; 197(8): 3086–3098]. The authors describe “TCR activation further promoted protein O-GlcNAcylation in Treg cells ex vivo (Figure 1C).” Were the cell lysates used for analysis of O-GlcNAc level by immunoblot in Figure 1C from sorted Treg cells or whole cell stimulated with anti-CD3/ CD28 beads? How many samples and how many independent

experiments have been done for immunoblot experiments? The protein markers (ladder) are missing in immunoblot results shown in Fig 1C and Supplement 1B.

Response: The original Fig. 1C was obtained from whole cells treated with anti-CD3/CD28 beads, which included non-Treg cells. Thus, we removed it in the revised manuscript. To specifically examine Treg cells, we repeated the experiment and gated on CD4⁺CD25⁺FOXP3⁺ Treg cells using flow cytometry. Three biological replicates were included in each group. We observed a significant upregulation of O-GlcNAcylation levels in Treg cells after CD3 and CD28 stimulation (Fig. 1C). Protein markers have also now been added in Supplemental Fig. 1B.

3. Figures 1J-1M show reduced MFI of foxp3 in CD4⁺CD25⁺ and CD4⁺CD25⁺ Treg frequency in Ubc-Cre/ERT2+Ogtl/Y cells treated with 4-OHT. What is the frequency of Treg cells identified with CD4⁺Foxp3⁺ or CD4⁺CD25⁺FOXP3⁺ cells?

Response: To precisely determine FOXP3 expression and Treg frequency, we re-analyzed the data by gating on the CD4⁺FOXP3⁺ population. As shown in the new Fig. 2B-D, MFI of FOXP3 was significantly reduced in CD4⁺FOXP3⁺ Treg cells and the frequency CD4⁺FOXP3⁺ Treg cells reduced from 16.2% to 6.2% after 4-OHT treatment.

4. IL4, IL13, and IL5 are produced by Th2 effector T cells. Differential expression levels of Th2 cytokines are seen in Th2 cells in vivo. A previous study suggested that increased GATA-3 expression is essential for IL-5 and IL-13 production by Th2 cells, but not IL-4 [Nat Immunol. 2004 Nov;5(11):1157-65]. The results in Figure 2J show increased GATA3 expression in Foxp3^{YFP-Cre/Y} Ogtl/Y mice. What would the IL-13 and IL-5 expression levels be in effector CD4⁺ T cells in Foxp3^{YFP-Cre/Y} Ogtl/Y mice, although slightly elevated IL4 expression has been shown in Figure 2L?

Response: We appreciate the reviewer's insight on differential expression of Th2 cytokines by Th2 cells. To determine IL-5 and IL-13 expression, we analyzed a new batch of mice (n = 5) and observed a mild but not significant increase of IL-5-expressing CD4⁺Foxp3⁻ T cells in Foxp3^{YFP-Cre/Y} Ogtl/Y mice (Supplementary Fig. 3H). We did not observe any difference of IL-13 between groups (Supplementary Fig. 3I). Even though GATA3⁺ Th2 cells increased their number in Foxp3^{YFP-Cre/Y} Ogtl/Y mice as a result of the loss of Treg suppressive function and/or the emergence of ex-Treg cells (Fig. 3H, J), Th2 cytokines only showed a tendency to increase, suggesting that Th1 inflammatory response is dominant in Foxp3^{YFP-Cre/Y} Ogtl/Y mice.

5. Figure 3I and 3J show the expression of Th1 T-bet in OGT-deficient ex-Treg cells. What would the expression level be in other Th cells such as Th17 TF RORγ and Th2 TF GATA3?

Response: Using the tamoxifen-inducible Foxp3^{eGFP-Cre-ERT2/Y} Ogtl/Y Rosa26^{tdTomato/wt} mice, we conclusively demonstrated the existence of ex-Treg cell (Fig. 4G). In addition, these ex-Treg cells were prone to express the Th1 transcription factor T-BET and the Th2 transcription factor GATA3 (Fig. 4H, I) but not RORγ, indicating the functional conversion of OGT-deficient Treg cells to Th-like phenotypes.

6. A previous study demonstrated the presence of many Treg IL-2 dependent genes, such as Klrp1, IL10, Blimp-1 et al [Immunity. 2009 Feb 20;30(2):204-17]. The results in Figure 5E-G show KLRG1 expression for evaluation the responsiveness of Treg cells to IL-2. It would be interesting to examine the other IL-2 dependent targets in Treg cells.

Response: We thank the reviewer for the critical suggestion. In order to profile the expression of other IL-2 target genes, we treated a new batch of female *Foxp3*^{YFP-Cre/wt} *Ogt*^{fl/fl} KO mice with the IL-2 complex. As expected, we found that the expansion of KLRG1⁺, GZMB⁺, and BLIMP-1⁺ eTreg populations was absent in OGT-deficient Treg cells, when compared to OGT-sufficient cells (Fig. 6E-J). Moreover, the increase in the expression of CD25 and IL-10 induced by IL-2 was inhibited when OGT was deleted (Supplementary Fig. 7C, D). These data strongly suggest that OGT is indispensable for proper IL-2 pathway activation in Treg cells.

7. Comparisons of MFI of *Foxp3* expression and CD25+FOXP3+ Treg frequency from *Foxp3*YFP-Cre/YOgtfl/YRosa26Stat5b-CA/wt mice (STAT5B-CA overexpression OGT-deficient Treg cells) and *Foxp3*YFP-Cre/YOgtfl/Y mice (OGT-deficient Treg cells) are shown in Figures 6D and 6E. To demonstrate the rescue dysfunction of OGT deficient Treg cells, it would be more explicit to include the Treg analysis from *Foxp3*YFP-Cre/YOgtwt/Y mice.

Response: Following the reviewer's suggestion, we have now included the *Foxp3*^{YFP-Cre/Y} *Ogt*^{wt/Y} control group in the revised Fig. 7. As shown in Fig 7D, STAT5B-CA overexpression rescued FOXP3 expression to a level comparable to wildtype controls. OGT deficiency did not change the frequency of Treg cells but STAT5B-CA was able to further increase Treg cell number (Fig. 7E). Even though Treg lineage instability in *Foxp3*^{YFP-Cre/Y} *Ogt*^{fl/Y} mice seemed to be fully restored by STAT5B-CA (Fig. 7H), Treg effector function was only partially rescued (Fig. 7G and Supplementary Fig. 9D).

8. For the suppression function of human Treg cells, in Figure 7E, please add the representative flow plots showing the cell proliferation of responder cells based on the dilution of fluorescence intensity of CFSE of gated cells.

Response: The representative flow plots have now been include in Supplementary Fig.10C.

Reviewer #3:

• While the authors report an impressive functional outcome from inhibition of OGT in Treg cells, the authors describe an incomplete molecular mechanism through FOXP3 and STAT5 and linkage of the TCR and IL-2 pathways. The connection of O-GlcNAc on FOXP3 to the phenotype observed in OGT-deficiency is tenuous at best. The authors find that FOXP3 is moderately stabilized by O-GlcNAc, but very limited connection of the glycosylation on FOXP3 is related back to the broader mechanistic outcome. Most of these studies rely on global manipulation of O-GlcNAc levels through genetic or chemical means. Knockdown of OGT will affect a number of functions including destabilization of FOXP3 as the authors show. The best approach to relate function back to the specific glycosites is to trace down the glycosites for mutation and correlate the mutants with reduction of the phenotype in line with global methods to regulate O-GlcNAc that the authors used. Otherwise, the molecular outcomes from OGT knockdown in Treg cells may truly derive from mechanisms through other proteins. Further examination of O-GlcNAc sites on FOXP3 would add to whether or not it plays a direct role in mediating auto-immune regulation in Treg cells.

Response: We of course agree that characterizing the glycosites on FOXP3 is of great interest. To identify sites of FOXP3 O-GlcNAcylation, Flag-tagged FOXP3 was expressed in HEK 293 cells together with OGT, immunopurified with anti-Flag beads, trypsin digested, and analyzed by liquid chromatography with tandem mass spectrometry (LC-MS/MS) using electron transfer dissociation (ETD). Multiple O-GlcNAcylation sites were identified (Supplemental Table 1).

Mutating 5 of these sites, including Thr38, Ser57, Ser58, Ser270, and Ser273, to alanine (5A) on the FOXP3 protein significantly blunted its O-GlcNAcylation level (Fig. 2E), reduced its stability (Fig. 2F), and ablated its transcriptional suppression activity induced by OGT (Supplementary Fig. 2F). We then retrovirally transduced FOXP3 and FOXP3-5A into CD4⁺CD25⁻ conventional T cells and flow cytometric analyses of transduced cells showed that the protein expression level of FOXP3-5A was much lower than that of wildtype FOXP3 (Fig. 2F). These results indicate that O-GlcNAcylation is required to stabilize FOXP3.

In the revised manuscript, we have also included further characterization of molecular mechanisms by which O-GlcNAcylation regulates FOXP3 and STAT5. Through RNA-seq, we identified IL-2/STAT5 signaling is a major affected pathway after loss of OGT in Treg cells. By treating female heterozygous mice with IL-2/anti-IL-2 antibody complexes, we found OGT-deficient cells were not responsive to IL-2 stimulation to expand IL-2/STAT5-target genes including KLRG1, GZMB, BLIMP-1, CD25, and IL-10 (Fig. 6E-J and Supplementary Fig. 7C, D). We then retrovirally expressed a constitutively active STAT5A (cS5) in Treg cells and observed increased expression of the STAT5-target genes *Socs1* and *Socs3*, whereas the O-GlcNAc-deficient cS5-T92A diminished such effect (Fig. 6L, M).

In isolated mouse Treg cells, treatment with an OGA inhibitor TMG, to elevate global protein O-GlcNAcylation, was able to increase the expression of FOXP3 protein and STAT5-target genes (Fig. 8A-F). In human Treg cells, OGA also promoted STAT5-target gene expression and the suppressive function (Fig. 8G-K). These data suggest that FOXP3 and STAT5 are two downstream targets of OGT. Nonetheless, we agree with the reviewer that a number of proteins/pathways/functions may be affected by OGT ablation in Treg cells. This is supported by the fact that STAT5-CA expression, which restored FOXP3 expression, only partially rescued the scurfy phenotype. Further investigations are warranted in the lab to examine O-GlcNAc-site-specific effect of FOXP3 and STAT5 in immune suppression in vivo, to systemically determine the O-GlcNAc proteome in Treg cells, and to study other potential molecular mechanisms mediating the regulation by protein O-GlcNAcylation.

- The authors do examine constitutively active STAT5 via a mutagenesis approach; however, these studies are uniformly performed in OGT-deficient mice, leading to a lack of clarity about STAT5 rescue being related to O-GlcNAc (what is the rationale for knocking out an O-GlcNAc site on STAT5 in O-GlcNAc deficient mice?). Nonetheless, the manuscript describes a surprising and intriguing dysregulation of Treg cells without O-GlcNAc that adds to the growing importance of O-GlcNAc in the immune system.

Response: The reviewer raised a critical question: to what extent the rescue by STAT5-CA was dependent on STAT5-O-GlcNAcylation. First to clarify, we did not knock out the O-GlcNAc site (T92) on STAT5 in *Foxp3^{YFP-Cre/Y} Ogt^{fl/Y}* KO mice. We expressed a constitutively active form of STAT5B (STAT5B-CA) by replacing histidine 299 and serine 711 with arginine and phenylalanine, respectively. This STAT5B-CA has been shown by many groups to be constitutively phosphorylated at tyrosine 699, mimicking the activation process^{19, 20, 21, 22}. STAT5B-CA expression specifically in Treg cells was able to bypass the defective O-GlcNAcylation, restored FOXP3 expression, prevented the emergence of ex-Treg cells, and partially ameliorated Treg-cell dysfunction and systemic inflammation (Fig. 7). In addition, in OGT-deficient Treg cells, overexpression of cS5 but not cS5-T92A was able to increase the expression of STAT5-target gene expression, suggesting that STAT5 O-GlcNAcylation is required for its transcriptional activity (Supplementary Fig. 8D). Nonetheless, conclusive evidence is still lacking to define the precise contribution of STAT5 O-GlcNAcylation in Treg cell function in vivo. Future development of new tools, such as STAT5-T92A knockin mice, are needed to fully answer this question.

- Rescue of mice at week 2 (Figure 6), how does it longitudinally correlate to these mice at week 4 and their longevity?

Response: As shown in Fig. 7C, STAT5B-CA rescue increased median life span from 32 days to 53 days, and maximum life span from 35 days to 108 days. At the age of 4 weeks, all *Foxp3*^{YFP-Cre/Y} *Ogt*^{fl/Y} KO mice are moribund (Fig. 3A), while *Foxp3*^{YFP-Cre/Y} *Ogt*^{fl/Y} *Rosa26*^{Stat5b-CA/wt} mice just started to show inflammation (data not shown). The fatal autoimmunity later observed in *Foxp3*^{YFP-Cre/Y} *Ogt*^{fl/Y} *Rosa26*^{Stat5b-CA/wt} mice suggests the existence of O-GlcNAc-regulated, non-STAT5 signaling pathways that contribute to Treg cell function.

- Is O-GlcNAcylation stabilizing FOXP3 after TCR activation or simply stabilizing it as it is being expressed more?

Response: During development, thymocytes that bear a moderately self-reactive TCR are selected to express FOXP3 and become Treg cells. Mature Treg cells also continuously receive self-reactive TCR signals, which are required to maintain the functional program of Treg cells. In Treg cells with TCR ablation, there was only a slight reduction in FOXP3 protein expression, without affecting *Foxp3* transcription^{23, 24}. The activation of the immune system in the case of Treg-specific TCR deletion mice was much milder than we observed in *Foxp3*^{YFP-Cre/Y} *Ogt*^{fl/Y} mice, suggesting that O-GlcNAcylation plays fundamental and constitutive roles in regulating Treg cells. Similar to findings in conventional T cells, we could detect high levels of OGT and protein O-GlcNAcylation in unstimulated Treg cells. TCR activation promoted protein O-GlcNAcylation, accompanied with upregulated FOXP3 expression (Fig.1A-C). We thus postulate that basal O-GlcNAcylation stabilizes FOXP3 and maintains the Treg cell lineage, while TCR-activated O-GlcNAcylation further strengthens such effects and also establishes a TCR-dependent O-GlcNAc profile to modulate immune suppression by Treg cells.

References:

1. Sun C, Shang J, Yao Y, Yin X, Liu M, Liu H, *et al.* O-GlcNAcylation: a bridge between glucose and cell differentiation. *Journal of Cellular and Molecular Medicine* 2016, **20**(5): 769-781.
2. Zeng H, Yang K, Cloer C, Neale G, Vogel P, Chi H. mTORC1 couples immune signals and metabolic programming to establish T(reg)-cell function. *Nature* 2013, **499**(7459): 485-490.
3. Kitagawa Y, Ohkura N, Kidani Y, Vandenbon A, Hirota K, Kawakami R, *et al.* Guidance of regulatory T cell development by Satb1-dependent super-enhancer establishment. *Nat Immunol* 2017, **18**(2): 173-183.
4. Dias S, D'Amico A, Cretney E, Liao Y, Tellier J, Bruggeman C, *et al.* Effector Regulatory T Cell Differentiation and Immune Homeostasis Depend on the Transcription Factor Myb. *Immunity* 2017, **46**(1): 78-91.
5. Xu M, Pokrovskii M, Ding Y, Yi R, Au C, Harrison OJ, *et al.* c-MAF-dependent regulatory T cells mediate immunological tolerance to a gut pathobiont. *Nature* 2018, **554**(7692): 373-377.

6. Cheng G, Yuan X, Tsai MS, Podack ER, Yu A, Malek TR. IL-2 receptor signaling is essential for the development of Klrp1+ terminally differentiated T regulatory cells. *J Immunol* 2012, **189**(4): 1780-1791.
7. Francisco LM, Salinas VH, Brown KE, Vanguri VK, Freeman GJ, Kuchroo VK, *et al.* PD-L1 regulates the development, maintenance, and function of induced regulatory T cells. *J Exp Med* 2009, **206**(13): 3015-3029.
8. Deaglio S, Dwyer KM, Gao W, Friedman D, Usheva A, Erat A, *et al.* Adenosine generation catalyzed by CD39 and CD73 expressed on regulatory T cells mediates immune suppression. *J Exp Med* 2007, **204**(6): 1257-1265.
9. Lehmann J, Huehn J, de la Rosa M, Maszyra F, Kretschmer U, Krenn V, *et al.* Expression of the integrin alpha Ebeta 7 identifies unique subsets of CD25+ as well as CD25- regulatory T cells. *Proc Natl Acad Sci U S A* 2002, **99**(20): 13031-13036.
10. Suffia I, Reckling SK, Salay G, Belkaid Y. A role for CD103 in the retention of CD4+CD25+ Treg and control of Leishmania major infection. *J Immunol* 2005, **174**(9): 5444-5455.
11. Cretney E, Xin A, Shi W, Minnich M, Masson F, Miasari M, *et al.* The transcription factors Blimp-1 and IRF4 jointly control the differentiation and function of effector regulatory T cells. *Nat Immunol* 2011, **12**(4): 304-311.
12. Freund P, Kerenyi MA, Hager M, Wagner T, Wingelhofer B, Pham HT, *et al.* O-GlcNAcylation of STAT5 controls tyrosine phosphorylation and oncogenic transcription in STAT5-dependent malignancies. *Leukemia* 2017.
13. Kearse KP, Hart GW. Lymphocyte activation induces rapid changes in nuclear and cytoplasmic glycoproteins. *Proc Natl Acad Sci U S A* 1991, **88**(5): 1701-1705.
14. Swamy M, Pathak S, Grzes KM, Damerow S, Sinclair LV, van Aalten DM, *et al.* Glucose and glutamine fuel protein O-GlcNAcylation to control T cell self-renewal and malignancy. *Nat Immunol* 2016, **17**(6): 712-720.
15. Golks A, Tran T-TT, Goetschy JF, Guerini D. Requirement for O-linked N-acetylglucosaminyltransferase in lymphocytes activation. *EMBO J* 2007, **26**(20): 4368-4379.
16. Lund PJ, Elias JE, Davis MM. Global Analysis of O-GlcNAc Glycoproteins in Activated Human T Cells. *J Immunol* 2016, **197**(8): 3086-3098.
17. Woo CM, Lund PJ, Huang AC, Davis MM, Bertozzi CR, Pitteri S. Mapping and quantification of over 2,000 O-linked glycopeptides in activated human T cells with isotope-targeted glycoproteomics (IsoTaG). *Mol Cell Proteomics* 2018.
18. Lopez Aguilar A, Gao Y, Hou X, Lauvau G, Yates JR, Wu P. Profiling of Protein O-GlcNAcylation in Murine CD8(+) Effector- and Memory-like T Cells. *ACS Chem Biol* 2017, **12**(12): 3031-3038.

19. Burchill MA, Goetz CA, Prlic M, O'Neil JJ, Harmon IR, Bensinger SJ, *et al.* Distinct effects of STAT5 activation on CD4+ and CD8+ T cell homeostasis: development of CD4+CD25+ regulatory T cells versus CD8+ memory T cells. *J Immunol* 2003, **171**(11): 5853-5864.
20. Onishi M, Nosaka T, Misawa K, Mui AL, Gorman D, McMahon M, *et al.* Identification and characterization of a constitutively active STAT5 mutant that promotes cell proliferation. *Mol Cell Biol* 1998, **18**(7): 3871-3879.
21. Burchill MA, Yang JY, Vogtenhuber C, Blazar BR, Farrar MA. IL-2 receptor beta-dependent STAT5 activation is required for the development of Foxp3(+) regulatory T cells. *Journal of Immunology* 2007, **178**(1): 280-290.
22. Chinen T, Kannan AK, Levine AG, Fan X, Klein U, Zheng Y, *et al.* An essential role for the IL-2 receptor in Treg cell function. *Nat Immunol* 2016, **17**(11): 1322-1333.
23. Levine AG, Arvey A, Jin W, Rudensky AY. Continuous requirement for the TCR in regulatory T cell function. *Nat Immunol* 2014, **15**(11): 1070-1078.
24. Vahl JC, Drees C, Heger K, Heink S, Fischer JC, Nedjic J, *et al.* Continuous T cell receptor signals maintain a functional regulatory T cell pool. *Immunity* 2014, **41**(5): 722-736.

REVIEWERS' COMMENTS:

Reviewer #1 (Remarks to the Author):

The authors have provided a very thorough and thoughtful response to my critiques. As a result, this is a stronger and more complete study that should generate significant interest in the field.

- Lawrence P. Kane, Professor of Immunology

Reviewer #2 (Remarks to the Author):

I appreciate the author's efforts to address reviewers' concerns. They have reanalyzed certain experiments (such as O-GlcNAcylation levels on gated CD4+CD25+FOXP3+ Treg cells and the frequency of CD4+FOXP3 Treg cells), removed the analysis of O-GlcNAc level on whole cells which include non-Treg cells, and performed additional experiments (for determining Th2 cytokines expression, expression of IL-2 target genes). In the revised manuscript the authors have addressed reviewer's concerns.

Reviewer #3 (Remarks to the Author):

The authors have responded to this reviewer's comments in a satisfactory manner. Only the addition of quantification for Figure 2F is requested as the levels of FOXP3 are difficult to distinguish.

Responses to Reviewers' comments

Reviewer #1 (Remarks to the Author):

The authors have provided a very thorough and thoughtful response to my critiques. As a result, this is a stronger and more complete study that should generate significant interest in the field.

- Lawrence P. Kane, Professor of Immunology

Response: Thank you very much Dr. Kane.

Reviewer #2 (Remarks to the Author):

I appreciate the author's efforts to address reviewers' concerns. They have reanalyzed certain experiments (such as O-GlcNAcylation levels on gated CD4+CD25+FOXP3+ Treg cells and the frequency of CD4+FOXP3 Treg cells), removed the analysis of O-GlcNAc level on whole cells which include non-Treg cells, and performed additional experiments (for determining Th2 cytokines expression, expression of IL-2 target genes). In the revised manuscript the authors have addressed reviewer's concerns.

Response: Thank you very much.

Reviewer #3 (Remarks to the Author):

The authors have responded to this reviewer's comments in a satisfactory manner. Only the addition of quantification for Figure 2F is requested as the levels of FOXP3 are difficult to distinguish.

Response: Thank you. Regarding Figure 2F, please focus on the 2h time point, when FOXP3-5A lost significantly more protein, compared to wildtype FOXP3. It supports the notion that loss of O-GlcNAcylation destabilizes FOXP3 protein.